# Detecting danger in gridworlds using Gromov's Link Condition

**Thomas F Burns**                                                         *thomas.burns@oist.jp*
*Neural Coding and Brain Computing Unit*
*OIST Graduate University*
*1919-1 Tancha, Onna-son, Kunigami-gun*
*Okinawa, Japan 904-0495*

**Robert Tang**                                                         *robert.tang@xjtlu.edu.cn*
*Department of Pure Mathematics*
*Xi'an Jiaotong–Liverpool University*
*111 Ren'ai Road, Suzhou Industrial Park*
*Suzhou, Jiangsu Province, China 215123*

**Reviewed on OpenReview:** *https://openreview.net/forum?id=t4p612DftO*

## Abstract

Gridworlds have been long-utilised in AI research, particularly in reinforcement learning, as they provide simple yet scalable models for many real-world applications such as robot navigation, emergent behaviour, and operations research. We initiate a study of gridworlds using the mathematical framework of *reconfigurable systems* and *state complexes* due to Abrams, Ghrist & Peterson. State complexes, a higher-dimensional analogue of state graphs, represent all possible configurations of a system as a single geometric space, thus making them conducive to study using geometric, topological, or combinatorial methods. The main contribution of this work is a modification to the original Abrams, Ghrist & Peterson setup which we introduce to capture agent braiding and thereby more naturally represent the topology of gridworlds. With this modification, the state complexes may exhibit geometric defects (failure of *Gromov's Link Condition*). Serendipitously, we discover these failures for agent-only cases occur exactly where undesirable or dangerous states appear in the gridworld. Our results therefore provide a novel method for seeking guaranteed safety limitations in discrete task environments with single or multiple agents, and offer useful safety information (in geometric and topological forms) for incorporation in or analysis of machine learning systems. More broadly, our work introduces tools from geometric group theory and combinatorics to the AI community and demonstrates a proof-of-concept for this geometric viewpoint of the task domain through the example of simple environments.

## 1 Introduction

The notion of a state (or configuration/phase) space is commonly used in mathematics and physics to represent all the possible states of a given system as a single geometric (or topological) object. This perspective provides a bridge which allows for tools from geometry and topology to be applied to the system of concern. Moreover, certain features of a given system are reflected by some geometric aspects of the associated state space (such as gravitational force being captured by *curvature* in spacetime). Thus, insights into the structure of the original system can be gleaned by reformulating them in geometric terms.

In discrete settings, state spaces have most commonly be represented in the artificial intelligence (AI) literature using graphs. In a *state graph*, each vertex represents a state, and states are connected via edges if a transition between them is dynamically possible. Such graphs are typically incorporated within Markov decision processes (MDPs), including in multi-agent navigation and cooperation tasks (Rizk et al., 2019). MDPs include additional information about probabilities of and expected rewards associated with particular

state transitions. Within this literature, there has been some work of a more geometric (Arslan et al., 2016) or topological (Waradpande et al., 2020) flavour. With few exceptions, however, where such works incorporate geometric realisations, concepts, or techniques, they typically do so in an empirical way and do not utilise higher dimensional objects such as simplicial or cube complexes.

In the current work, we focus on a higher dimensional analogue of state graphs which take the form of cube complexes. Abrams, Ghrist & Peterson's *state complexes* (Abrams & Ghrist, 2004; Ghrist & Peterson, 2007) provide a general framework for representing discrete reconfigurable systems as non-positively curved (NPC) cube complexes, giving access to a wealth of mathematical and computational benefits via efficient optimisation algorithms guided by geometric insight (Ardila et al., 2012). These have been used to develop efficient algorithms for robotic motion planning (Ardila et al., 2014; 2017) and self-reconfiguration of modular robots (Larkworthy & Ramamoorthy, 2010). NPC cube complexes also possess rich hyperplane structures which geometrically capture binary classification (Chatterji & Niblo, 2005; Wise, 2012; Sageev, 2014). However, their broader utility to fields like AI has until now been relatively unexplored.

Our main contribution is the first application of this geometric approach (of using state complexes) to the setting of multi-agent gridworlds. We introduce a natural modification to the state complex appropriate to the setting of multi-agent gridworlds (to capture the braiding or relative movements of agents); however, in the agent-only case this can lead to state complexes which are no longer NPC. Nevertheless, by applying Gromov's Link Condition, we completely characterise when positive curvature occurs in our new state complexes for these agent-only cases, and relate this to features of the gridworlds (see Theorem 5.2). Serendipitously, we discover that the states where Gromov's Link Condition fails are those in which agents can potentially collide. In other words, collision-detection is naturally embedded into the intrinsic geometry of the system via our modification. Current approaches to collision-detection and navigation during multi-agent navigation often rely on modelling and predicting collisions based on large training datasets (Kenton et al., 2019; Fan et al., 2020; Qin et al., 2021) or by explicitly modelling physical movements (Kano et al., 2021). However, our approach is purely geometric, requires no training, and can accommodate many conceivable types of actions and inter-actions, not just simple movements.

In the context of single-agent motor learning and navigation, there has been some work (Peng & van de Panne, 2017; Reda et al., 2020; Schneider et al., 2023) on how the choices of what and how information is represented in a learning system can effect task performance. Implicitly, these works can be viewed as empirical investigations on how changes to the geometry or topology of state spaces relate to the efficacy of the applied learning algorithms. However, these studies do not seek to incorporate higher-level geometric or topological information of the originating domain or task in a substantial way, or investigate this formally, before applying or investigating the performance of the learning algorithms – and even fewer do so for multi-agent systems (Rizk et al., 2019). One possible reason for this is a lack of known suitable tools. Our experimental and theoretical results show there is a wealth of geometric information available in (even very simple) task domains, which is accessible using tools from geometric group theory and combinatorics. Our work therefore joins a growing body of research aimed towards understanding AI systems from a more geometric perspective (Hauser & Ray, 2017; Lei et al., 2020; Stephenson et al., 2021; Archer et al., 2021; Stober et al., 2011).

## 2 State complex of a gridworld

A *gridworld* is a two-dimensional, flat array of *cells* arranged in a grid, much like a chess or checker board. Each cell can be occupied or unoccupied. A cell may be occupied, in our setting, by one and only one freely-moving agent or movable object. Other gridworlds may include rewards, punishments, buttons, doors, locks, keys, checkpoints, dropbears, etc., much like many basic video games. Gridworlds have been a long-utilised setting in AI research, particularly reinforcement learning, since they are simple yet scalable in size and sophistication (Da Silva et al., 2020; Waradpande et al., 2020). They also offer clear analogies to many real-world applications or questions, such as robot navigation (Hodge et al., 2021), emergent behaviour (Kajic et al., 2020), and operations research (Laurent et al., 2021). For these reasons, gridworlds have also been developed for formally specifying problems in AI safety (Leike et al., 2017).

A *state* of a gridworld can be encoded by assigning each cell a *label*. In the example shown in Figure 1, these labels are shown for an agent, an object, and empty floor. A change in the state, such as an agent moving from one cell to an adjacent empty cell, can be encoded by *relabelling* the cells involved. This perspective allows us to take advantage of the notion of *reconfigurable systems* as introduced by Abrams, Ghrist & Peterson (Abrams & Ghrist, 2004; Ghrist & Peterson, 2007).

More formally, consider a graph $G$ and a set $\mathcal{A}$ of labels. A *state* is a function $s : V(G) \to \mathcal{A}$, i.e. an assignment of a label to each vertex of $G$. A possible relabelling is encoded using a *generator* $\phi$; this comprises the following data:

Figure 1: A $3 \times 3$ gridworld with one agent (a koala) and one object (a beach ball).

- a subgraph $SUP(\phi) \subseteq G$ called the *support*;

- a subgraph $TR(\phi) \subseteq SUP(\phi)$ called the *trace*; and

- an unordered pair of *local states*

$$u_0^{loc}, u_1^{loc} : V(SUP(\phi)) \to \mathcal{A}$$

that agree on $V(SUP(\phi)) - V(TR(\phi))$ but differ on $V(TR(\phi))$.

A generator $\phi$ is *admissible* at a state $s$ if $s|_{SUP(\phi)} = u_0^{loc}$ (or $u_1^{loc}$), in other words, if the assignment of labels to $V(SUP(\phi))$ given by $s$ completely matches the labelling from (exactly) one of the two local states. If this holds, we may apply $\phi$ to the state $s$ to obtain a new state $\phi[s]$ given by

$$\phi[s](v) := \begin{cases} u_1^{loc}(v), & v \in V(TR(\phi)) \\ s(v), & \text{otherwise.} \end{cases}$$

This has the effect of relabelling the vertices in (and only in) $TR(\phi)$ to match the other local state of $\phi$. Since the local states are unordered, if $\phi$ is admissible at $s$ then it is also admissible at $\phi[s]$; moreover, $\phi[\phi[s]] = s$.

**Definition 2.1** (Reconfigurable system (Abrams & Ghrist, 2004; Ghrist & Peterson, 2007)). A *reconfigurable system* on a graph $G$ with a set of labels $\mathcal{A}$ consists of a set of generators together with a set of states closed under the action of admissible generators.

Configurations and their reconfigurations can be used to construct a *state graph* (or transition graph), which represents all possible states and transitions between these states in a reconfigurable system. More formally:

**Definition 2.2** (State graph). The state graph $\mathcal{S}^{(1)}$ associated to a reconfigurable system has as its vertices the set of all states, with edges connecting pairs of states differing by a single generator.[1]

Let us now return our attention to gridworlds. We define a graph $G$ to have vertices corresponding to the cells of a gridworld, with two vertices declared adjacent in $G$ exactly when they correspond to neighbouring cells (i.e. they share a common side). Our set of labels is chosen to be

$$\mathcal{A} = \{\text{`agent'}, \text{`object'}, \text{`floor'}\}.$$

We do not distinguish between multiple instances of the same label. We consider two generators:

- **Push/Pull.** An agent adjacent to an object is allowed to push/pull the object if there is an unoccupied floor cell straight in front of the object/straight behind the agent; and

- **Move.** An agent is allowed to move to a neighbouring unoccupied floor cell.

---

[1]For a cell complex $\mathcal{X}$, it is standard notation to use $\mathcal{X}^{(k)}$ for its $k$-skeleton (the subcomplex formed by the union of all cells of dimension at most $k$).

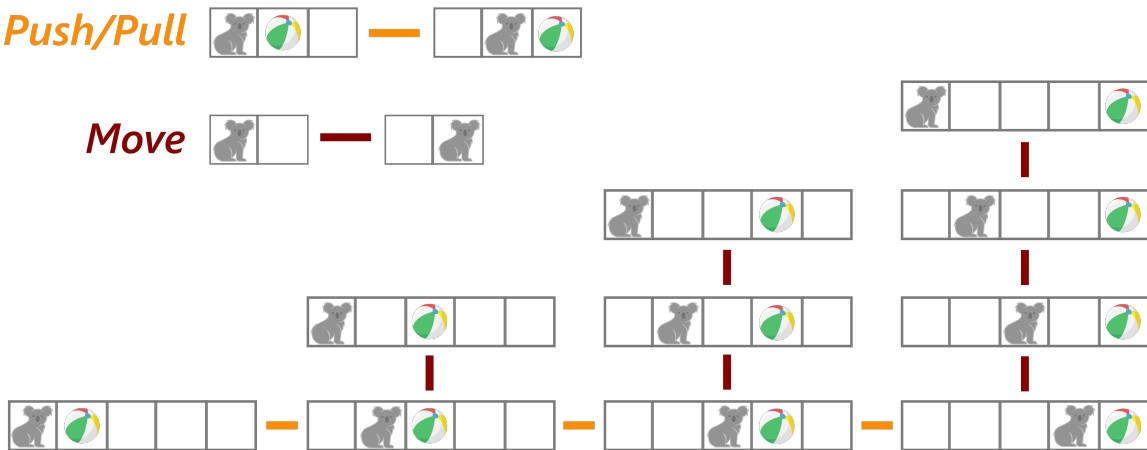

Figure 2: An example $1 \times 5$ gridworld with one agent and one object with two generators – Push/Pull and Move – and the resulting state graph. In the state graph, edge colours indicate the generator type which relabels the gridworld.

These two generators have the effect of enabling agents to at any time move in any direction not blocked by objects or other agents, and for agents to push or pull objects within the environment into any configuration if there is sufficient room to move. For both types of generators, the trace coincides with the support. For the Push/Pull generator, the support is a row or column of three contiguous cells, whereas for the Move generator, the support is a pair of neighbouring cells. A simple example of a state graph, together with the local states for the two generator types, is shown in Figure 2.

In a typical reconfigurable system, there may be many admissible generators at a given state $s$. If the trace of an admissible generator $\phi_1$ is disjoint from the support of another admissible generator $\phi_2$, then $\phi_2$ remains admissible at $\phi_1[s]$. This is because the relabelling by $\phi_1$ does not interfere with the labels on $SUP(\phi_2)$. More generally, a set of admissible generators $\{\phi_1, \ldots, \phi_n\}$ at a state $s$ *commutes* if $SUP(\phi_i) \cap TR(\phi_j) = \emptyset$ for all $i \neq j$. When this holds, these generators can be applied independently of one another, and the resulting state does not depend on the order in which they are applied. A simple example of this in the context of gridworlds is a large room with $n$ agents spread sufficiently far apart to allow for independent simultaneous movement.

Abrams, Ghrist & Peterson represent this mutual commutativity by adding higher dimensional cubes to the state graph to form a cube complex called the *state complex*. We give an informal definition here, and refer to their papers for the precise formulation (Abrams & Ghrist, 2004; Ghrist & Peterson, 2007).

Cube complexes are higher dimensional analogues of graphs that appear prominently in topology, geometric group theory, and combinatorics. Background on cube complexes can be found in Schwer (2019); Sageev (2014); Wise (2012)[2].

Informally, a cube complex is a space that can be constructed by gluing cubes together in a fashion not too dissimilar to a child's building blocks. An $n$–cube is modelled on

$$\{(x_1, \ldots, x_n) \in \mathbb{R}^n \ : 0 \leq x_i \leq 1 \text{ for all } i\}.$$

By restricting some co-ordinates to either 0 or 1, we can obtain lower dimensional subcubes. In particular, an $n$–cube has $2^n$ vertices and is bounded by $2n$ faces which are themselves $(n-1)$–cubes. A *cube complex $X$* is a union of cubes, where the intersection of every pair of distinct cubes is either empty, or a common subcube.

In the state complex, if $\{\phi_1, \ldots, \phi_n\}$ is a set of commuting admissible generators at a state $s$ then there are $2^n$ states that can be obtained by applying any subset of these generators to $s$. These $2^n$ states form

---

[2]Much of the literature in geometric group theory focusses primarily on non-positively curved cube complexes, whereas in our study, the presence of positive curvature plays a crucial role.

the vertices of an $n$–cube in the state complex. Each $n$–cube is bounded by $2n$ faces, where each face is an $(n-1)$–cube: by disallowing a generator $\phi_i$, we obtain a pair of faces corresponding to those states (in the given $n$–cube) that agree with one of the two respective local states of $\phi_i$ on $SUP(\phi_i)$.

**Definition 2.3** (State complex)**.** The *state complex* $\mathcal{S}$ of a reconfigurable system is the cube complex constructed from the state graph $\mathcal{S}^{(1)}$ by inductively adding cubes as follows: whenever there is a set of $2^n$ states related by a set of $n$ admissible commuting generators, we add an $n$–cube so that its vertices correspond to the given states, and so that its $2n$ boundary faces are identified with all the possible $(n-1)$–cubes obtained by disallowing a generator. In particular, every cube is uniquely determined by its vertices.

In our gridworlds setting, each generator involves exactly one agent. This means commuting generators can only occur if there are multiple agents. A simple example of a state complex for two agents in a $2 \times 2$ room is shown in Figure 3. Note that there are six embedded 4–cycles in the state graph, however, only two of these are filled in by squares: these correspond to independent movements of the agents, either both horizontally or both vertically.

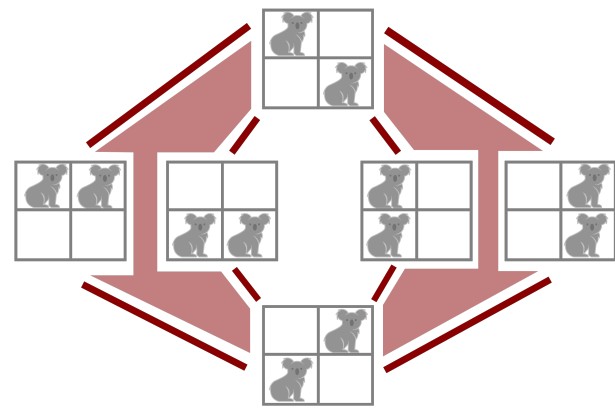

Figure 3: State complex of a $2 \times 2$ gridworld with two agents. Shading indicates squares attached to the surrounding 4–cycles.

## 3 Exploring gridworlds with state complexes

To compute the state complex of a (finite) gridworld, we first initialise an empty graph $\mathcal{G}$ and an empty 'to-do' list $\mathcal{L}$. As input, we take a chosen state of the gridworld to form the first vertex of $\mathcal{G}$ and also the first entry on $\mathcal{L}$. The state complex is computed according to a breadth-first search by repeatedly applying the following:

- Let $v$ be the first entry on $\mathcal{L}$. List all admissible generators at $v$. For each such generator $\phi$:
  - If $\phi[v]$ already appears as a vertex of $\mathcal{G}$, add an edge between $v$ and $\phi[v]$ (if it does not already exist).
  - If $\phi[v]$ does not appear in $\mathcal{G}$, add it as a new vertex to $\mathcal{G}$ and add an edge connecting it to $v$. Append $\phi[v]$ to the end of $\mathcal{L}$.
- Remove $v$ from $\mathcal{L}$.

The process terminates when $\mathcal{L}$ is empty. The output is the graph $\mathcal{G}$. When $\mathcal{L}$ is empty, we have fully explored all possible states that can be reached from the initial state. It may be possible that the true state graph is disconnected, in which case the above algorithm will only return a connected component $\mathcal{G}$. For our purposes, we shall limit our study to systems with connected state graphs[3]. From the state graph, we construct the state complex by first finding all 4–cycles in the state graph. Then, by examining the states involved, we can determine whether a given 4–cycle bounds a square representing a pair of commuting moves.

To visualise the state complex, we first draw the state graph using the Kamada–Kawai force-directed algorithm (Kamada & Kawai, 1989) which attempts to draw edges to have similar length. We then shade the region(s) enclosed by 4–cycles representing commuting moves. For ease of visual interpretation in our figures, we do

---

[3]There is no theoretical reason for limiting our study to connected state complexes. In fact, our main theorem would still hold without the connectedness assumption as it is a purely local statement. If one begins with an initial state, then its connected component comprises precisely all states which can be reached by applying some sequence of generators. Thus, from a practical and computational standpoint, one only ever deals with a connected component of the full state complex when exploring from a specific instance of a gridworld. Another issue is that without the connectedness assumption, the problem of finding a shortest path between two given states could be impossible.

not also shade higher-dimensional cubes, although such cubes are noticeable and can be easily computed and visualised if desired.

Constructing and analysing state complexes of gridworlds is in and of itself an interesting and useful way of exploring their intrinsic geometry. For example, Figure 4 shows the state complex of a $3 \times 3$ gridworld with one agent and one object. The state complex reveals two scales of geometry: larger 'blobs' of states organised in a $3 \times 3$ grid, representing the location of the object; and, within each blob, copies of the room's remaining empty space, in which the agent may walk around and approach the object to Push/Pull. Each 12–cycle 'petal' represents a 12–step choreography wherein the agent pushes and pulls the object around in a 4–cycle in the gridworld. In this example, the state complex is the state graph, since there are no possible commuting moves.

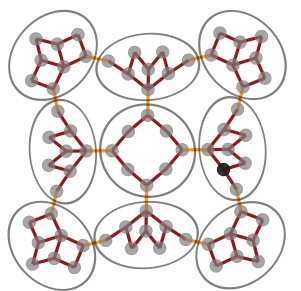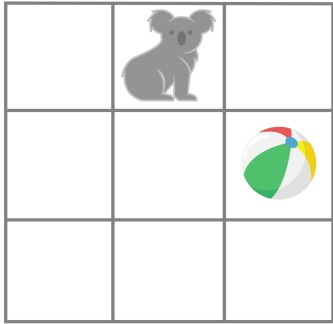

Figure 4: State complex (left) of a $3 \times 3$ gridworld with one agent and one object (right). The darker vertex in the state complex represents the state shown in the gridworld state on the right. Edges in the state complex are coloured according to their generator – orange for Push/Pull and maroon for Move. Grey circles which group states where the ball is static have been added to illustrate the different scales of geometry.

The examples discussed thus far all have planar state graphs. Planarity does not hold in general – indeed, the $n$–cube graph for $n \geq 4$ is non-planar, and a state graph can contain $n$–cubes if the gridworld has $n$ agents and sufficient space to move around. It is tempting to think that the state complex of a gridworld with more agents should therefore look quite different to one with fewer agents. However, Figure 5 shows this may not always be the case: there is a symmetry induced by swapping all 'agent' labels with 'floor' labels.

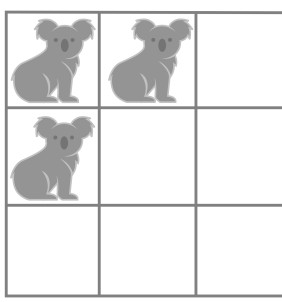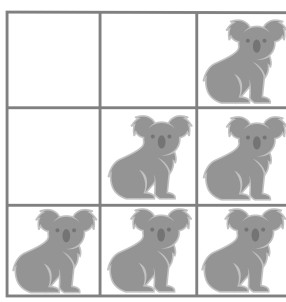

Figure 5: State complex (centre) of a $3 \times 3$ gridworld with three agents (left) and six agents (right). They share the same state complex due to the 'agent' $\leftrightarrow$ 'floor' label inversion symmetry.

We observe an interesting pattern in Figure 5, where 'clusters' of vertices in the state complex are arranged in a gridlike fashion on the plane. This gridlike arrangement is also exhibited in Figures 10 and 12 in Appendix A.1, as well as in other experiments involving agent-only gridworlds in larger rooms. It is worth emphasising that these visualisations were generated purely by feeding the state graph into the graph-drawing algorithm, with no human input involved. This suggests that the gridlike clustering could be revealing an intrinsic feature of the state complex, rather than simply being an artefact of a particular visualisation.

Indeed, by inspecting several examples, it appears that the visualisation approximates a natural projection map from the state complex of an agent-only gridworld to the integer lattice in $\mathbb{R}^2$. Let us identify the cell in row $i$ and column $j$ in a gridworld with the integer co-ordinate $(i, j) \in \mathbb{Z}^2$. Then for each state $s \in \mathcal{S}^{(0)}$,

define $f(s) \in \mathbb{Z}^2$ to be the sum of all co-ordinates corresponding to the cells occupied by an agent. We can view $f$ as a map which assigns a state to a 'cluster' in $\mathbb{Z}^2$. Applying a single Move generator to $s$ results in a state $s'$ such that $f(s')$ differs from $f(s)$ by either $(\pm 1, 0)$ or $(0, \pm 1)$. In particular, any pair of states mapping to a common 'cluster' in $\mathbb{Z}^2$ must be non-adjacent in the state graph. Thus, $f$ extends to a graph map from the state graph $\mathcal{S}^{(1)}$ to the standard integer grid (regarded as a graph on the plane).

It would be interesting to see if there could be other graph maps from the state graph to the integer grid which are not related to the map $f$ by an obvious symmetry. If the answer is no, then this could help explain why the graph-drawing algorithm appears to produce consistently similar projections for varying numbers of agents and sizes of rooms.

## 4  Dancing with myself

The state complex of a gridworld with $n$ agents can be thought of as a discrete analogue of the configuration space of $n$ points on the 2D–plane. However, there is a problem with this analogy: there can be 'holes' created by 4–cycles in the state complex where a single agent walks in a small square-shaped dance by itself, as shown in Figure 6.

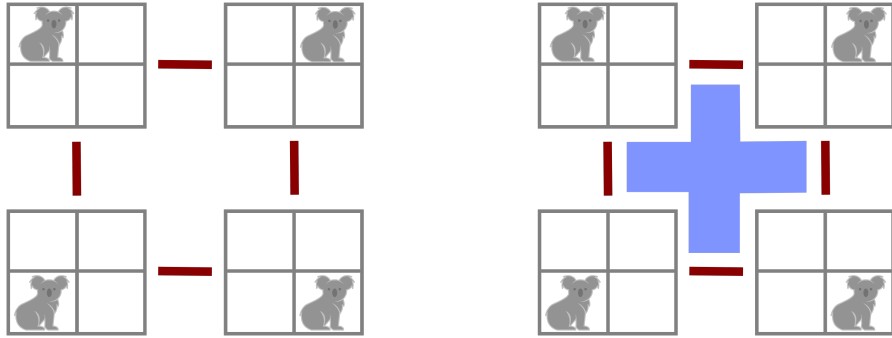

Figure 6: State complex of a $2 \times 2$ gridworld with one agent under the original definition of Abrams, Ghrist & Peterson (Abrams & Ghrist, 2004; Ghrist & Peterson, 2007) (left) and with our modification (right). The blue shading is a filled in square indicating a *dance*.

The presence of these holes would suggest something meaningful about the underlying gridworld's intrinsic topology, e.g., something obstructing the agent's movement at that location in the gridworld that the agent must move around. In reality, the environment is essentially a (discretised) 2D–plane with nothing blocking the agent from traversing those locations. Indeed, these 'holes' are uninteresting topological quirks which arise due to the representation of the gridworld as a graph. We therefore deviate from the original definition of state complexes by Abrams, Ghrist & Peterson (Abrams & Ghrist, 2004; Ghrist & Peterson, 2007) and choose to fill in these 'dance' 4–cycles with squares.[4]

Formally, we define a **dance** $\delta$ to comprise the following data:

- the support $SUP(\delta)$ given by a $2 \times 2$ subgrid in the gridworld,

- four local states defined on $SUP(\delta)$, each consisting of exactly one agent label and three floor labels, and

- four Move generators, each of which transitions between two of the four local states (as in Figure 6).

---

[4]Ghrist and Peterson themselves ask if there could be better ways to complete the state graph to a higher-dimensional object with better properties (Question 6.4 in Ghrist & Peterson (2007)).

We say that $\delta$ is *admissible* at a state $s$ if $s|_{SUP(\delta)}$ agrees with one of the four local states of $\delta$. Moreover, these four local states are precisely the states that can be reached when we apply some combination of the four constituent Moves. We do not define the trace of a dance, however, we may view the trace of each of the four constituent Moves as subgraphs of $SUP(\delta)$.

A dance is itself not a generator in the sense of Abrams, Ghrist & Peterson; rather, it is a set of 4 states related by 4 Move generators. Observe that a dance exists precisely when an agent is able to execute a 'diagonal move' via two different move sequences: a horizontal move followed by a vertical move; or a vertical followed by a horizontal one. We can thus view a dance as capturing a 'spatially commuting' relationship between these two move sequences. Note that these Moves are not commuting in the strict Abrams, Ghrist & Peterson sense (which could be viewed as a 'temporally commuting' relationship between generators), as the pairs of generators involved do not even coincide. Our modifications to the state complex, detailed below, allow us to treat both notions of commutativity simultaneously via cubes. This demonstrates a conceptual advantage of working with a complex rather than a graph: with a graph, we can only capture relations between pairs of states; by allowing higher-dimensional cubes, we can also capture relations between sets of generators or states.

The notion of commutativity can be extended to incorporate dancing as follows. Suppose that we have a set $\{\phi_1, \ldots, \phi_l, \delta_1, \ldots, \delta_m\}$ of $l$ admissible generators and $m$ admissible dances at a state $s$. We say that this set *commutes* if the supports of its elements are pairwise disjoint. When this holds, there are $2^{l+2m}$ possible states that can be obtained by applying some combination of the generators and dances to $s$: there are two choices of local state for each $\phi_i$, and four for each $\delta_j$. We capture this extended notion of commutativity by attaching additional cubes to the state complex to form our modified state complex.

**Definition 4.1** (Modified state complex). The *modified state complex* $\mathcal{S}'$ of a gridworld is the cube complex obtained by filling in the state graph $\mathcal{S}^{(1)}$ with higher dimensional cubes whenever there is a set of commuting moves or dances. Specifically, whenever a set of $2^{l+2m}$ states are related by a commuting set of $l$ generators and $m$ dances, we add an $n$–cube having the given set of states as its vertices, where $n = l + 2m$. Each of the $2n$ faces of such an $n$–cube is identified with an $(n-1)$–cube obtained by either disallowing a generator $\phi_i$ and choosing one of its two local states, or replacing a dance $\delta_j$ with one of its four constituent Moves.

Our modification removes uninteresting topology. This can be observed by examining 4–cycles in $\mathcal{S}'$. On the one hand, some 4–cycles are trivial (they can be 'filled in'): *dancing-with-myself* 4–cycles, and *commuting moves* (two agents moving back and forth) 4–cycles (which were trivial under the original definition). These represent trivial movements of agents relative to one another. On the other hand, there is a non-trivial 4–cycle in the state complex for two agents in a $2 \times 2$ room, as can be seen in the centre of Figure 3 (here, no dancing is possible so the modified state complex is the same as the original). This 4–cycle represents the two agents moving half a 'revolution' relative to one another – indeed, performing this twice would give a full revolution. (There are three other non-trivial 4–cycles, topologically equivalent to this central one, that also achieve the half-revolution.)

In a more topological sense[5], by filling in such squares and higher dimensional cubes, our state complexes capture the non-trivial, essential relative movements of the agents. This can be used to study the braiding or mixing of agents, and also allows us to consider path-homotopic paths as 'essentially' the same. One immediate difference this creates with the original state complexes is a loss of symmetries like those shown in Figure 5, since there is no label inversion for a dance when other agents are crowding the dance-floor.

*Remark* 4.2. Another approach could be to add edges to the state complex to represent diagonal moves instead of filling in 'dance' squares. However, doing so would further complicate the topology of the state complex. Heuristically, adding more edges would increase the rank of the fundamental group, introducing even more unnatural topology. In the example of a $2 \times 2$ gridworld with one agent, we would obtain a complete graph on 4 vertices as the state graph (or complex), whose fundamental group is a free group of rank 3. However, the original state complex is homotopy equivalent to a circle, with the group of integers (the free group of rank 1) as its fundamental group. Thus, we choose the approach of attaching higher dimensional cells in order to remove undesirable topology and reduce the rank. In the $2 \times 2$ example, the modified state complex is contractible and hence has trivial fundamental group.

---

[5]By considering the fundamental group.

## 5   Gromov's Link Condition

The central geometric characteristic of Abrams, Ghrist, & Peterson's state complexes is that they are *non-positively curved* (NPC). Indeed, this local geometric condition is conducive for developing efficient algorithms for computing geodesics. However, with our modified state complexes, this NPC geometry is no longer guaranteed – we test for this on a vertex-by-vertex basis using a classical geometric result due to Gromov (see also Theorem 5.20 of Bridson & Haefliger (1999) and Sageev (2014)).

Before discussing Gromov's Link Condition, we provide brief explanations of simplicial complexes and links. Readers familiar with these objects and concepts may safely skip. Further background on simplicial complexes are detailed in standard algebraic topology texts (Hatcher, 2002; Edelsbrunner & Harer, 2010).

**Simplicial complexes.**   Simplicial complexes are constructed in a similar manner to cube complexes, except that we use higher dimensional analogues of triangles or tetrahedra instead of cubes. An $n$–dimensional simplex (or $n$–simplex) is modelled on

$$\{(x_1, \ldots, x_{n+1}) \in \mathbb{R}^{n+1} \ : x_i \geq 0 \text{ for all } i, \ \sum_i x_i = 1\};$$

this has $n+1$ vertices and is bounded by $n+1$ faces which are themselves $(n-1)$–simplices. For $n = 0, 1, 2, 3$, an $n$–simplex is respectively a point, line segment, triangle, and tetrahedron. A *simplicial complex $K$* is an object that can be constructed by taking a graph and then inductively filling in simplices of progressively higher dimension; this graph is called the *1–skeleton* of $K$. We require that every finite set of vertices in $K$ form the vertices of (or *spans*) at most one simplex; thus simplices in $K$ are uniquely determined by their vertices. (This rules out loops or multi-edges in the 1–skeleton.)

**Links.**   The local geometry about a vertex $v$ in a cube complex $X$ is captured by a simplicial complex known as its *link $lk(v)$*. Intuitively, this is the intersection of a small sphere centred at $v$ within $X$, and can be regarded as the space of possible directions emanating from $v$. Each edge in $X$ emanating from $v$ determines a vertex (0–simplex) in $lk(v)$. If two such edges bound a 'corner' of a square in $X$ based at $v$, then there is an edge (1–simplex) connecting the associated vertices in $lk(v)$. More generally, each 'corner' of an $n$–cube incident to $v$ gives rise to an $(n-1)$–simplex in $lk(v)$; moreover, the boundary faces of the simplex naturally correspond to the faces of the cube bounding the corner. Since the cube complexes we consider have cubes completely determined by their vertices, each simplex in $lk(v)$ is also completely determined by its vertices. Figure 7 illustrates four separate examples of links of vertices in cube complexes.

**Gromov's Link Condition.**   Local curvature in a cube complex can be detected by examining the combinatorial structure of the links of its vertices. Specifically, Gromov's Link Condition gives a method for proving that a cube complex is NPC[6], where there is an absence of positive curvature. In the bottom-right example in Figure 7, where there is positive curvature, we observe a 'hollow' triangle in its link. In the other examples of Figure 7, where there is only negative or zero curvature, there are no such hollow triangles (or hollow simplices).

This absence of 'hollow' or 'empty' simplices is formalised by the *flag* property: a simplicial complex is *flag* if whenever a set of $n+1$ vertices spans a complete subgraph in the 1–skeleton, they must span an $n$–simplex. In particular, a flag simplicial complex is determined completely by its 1–skeleton. If $v$ is a vertex in a cube complex $X$, then the flag condition on $lk(v)$ can be re-interpreted as a 'no empty corner' condition for the cube complex: whenever we see (what appears to be) the corner of an $n$–cube, then the whole $n$–cube actually exists.

**Theorem** (Gromov's Link Condition (Gromov, 1987)). *A finite-dimensional cube complex $X$ is non-positively curved if and only if the link of each vertex in $X$ is a flag simplicial complex.* □

Thus, the local geometry of a cube complex is determined by the combinatorics of its links.

---

[6]In the sense that geodesic triangles are no fatter than Euclidean triangles (Bridson & Haefliger, 1999).

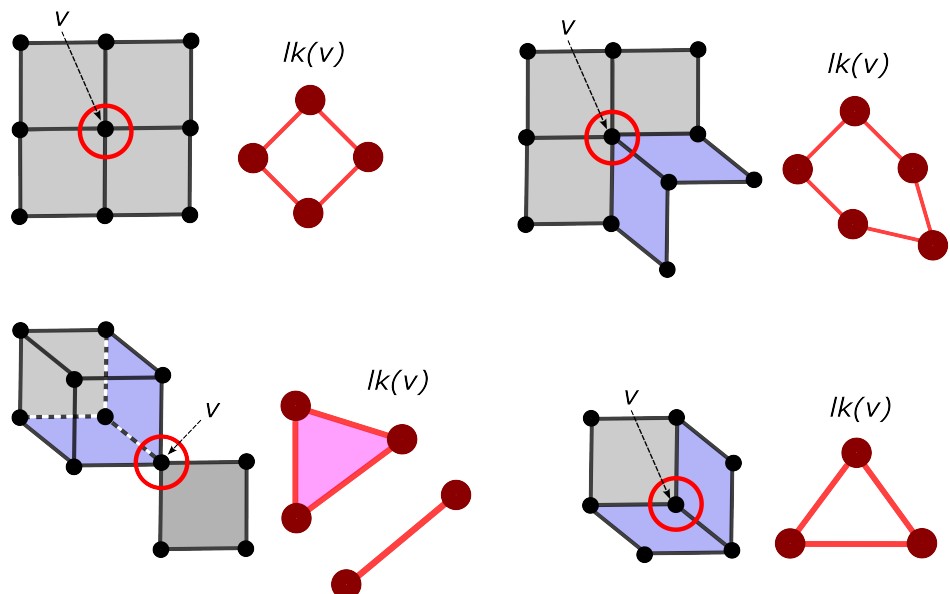

Figure 7: Four separate examples of links of vertices in cube complexes. In the bottom-right example, where there is positive curvature, $lk(v)$ is a 'hollow' triangle and is thus not a flag simplicial complex. For the other examples, $lk(v)$ is a flag complex and therefore, by Gromov's Link Condition, there is only negative or zero curvature. In the bottom-left example, the cube complex is a solid cube joined to a filled-in square at a common vertex $v$.

Under the Abrams, Ghrist & Peterson setup, if $v$ is a state in $\mathcal{S}$ then the vertices of its link $lk(v)$ represent the possible admissible generators at $v$. Since cubes in $\mathcal{S}$ are associated with commuting sets of generators, each simplex in $lk(v)$ represents a set of commuting generators. Gromov's Link Condition for $lk(v)$ can be reinterpreted as follows: whenever a set of admissible generators is *pairwise* commutative, then it is *setwise* commutative. Using this, it is straightforward for Abrams, Ghrist & Peterson to verify that this always holds for their state complexes (see Theorem 4.4 of Ghrist & Peterson (2007)).

*Remark* 5.1. The terms "non-positively curved (NPC)" and "locally CAT(0)" are synonymous in the context of geodesic metric spaces. For a cube complex, the CAT(0) property is equivalent to it being both NPC (via Gromov's Link Condition) and simply connected (i.e. it has trivial fundamental group). The state complexes of Abrams, Ghrist & Peterson are always NPC, but they are not necessarily simply connected. For example, the $2 \times 2$ gridworld with 2 agents is homotopy equivalent to a circle (see Figure 3), hence it is not simply connected and therefore not CAT(0). In the work of Ardila et al. (2012) on state complexes for robotic arms, it is the simple connectedness (a global property) which is harder to establish – NPC geometry holds immediately thanks to Abrams & Ghrist (2004); Ghrist & Peterson (2007). Thus, in the interest of technical correctness, we consider it preferable to use the terms "NPC" or "locally CAT(0)" instead of "CAT(0)" to describe the geometry of a cube complex when the status of simple connectedness is unknown.

For our modified states complexes, the situation is not as straightforward. The key issue is that our cubes do not only arise from commuting generators – we must take dances into account. Indeed, when attempting to prove that Gromov's Link Condition holds, we discovered some very simple gridworlds where it actually fails; see Figure 8 and Appendix A.1. Despite this apparent drawback, we nevertheless show that Figure 8 accounts for all the possible failures of Gromov's Link Condition in the setting of agent-only gridworlds[7].

---

[7]While writing this paper, the first author was involved in two scooter accidents – collisions involving only agents (luckily without serious injury). So, while this class of gridworlds is strictly smaller than those also involving objects or other labels, it is by no means an unimportant one. If only the scooters had Gromov's Link Condition checkers!

Failure of the Link Condition can indicate available moves at some state that cannot be safely performed simultaneously and independently without risking collisions between labels. Another interpretation of positive curvature in this context is something akin to what real-time computer strategy games call 'fog of war' (distance-dependent limiting of observations which extends from the player-controlled agents), and more specifically the viewable distance from an agent's line-of-sight. Such fog makes AI systems operating in such environments particularly challenging, although remarkable success has been achieved in games like StarCraft (Vinyals et al., 2019).

To further explicate the connection between the structure of links and collision avoidance in the gridworlds setting, consider the following setup. Each time step involves a *planning* phase and an *execution* phase. In the planning phase, each agent may propose an action (e.g., Move). Our modified setup also permits an agent to simultaneously propose both a horizontal and vertical Move from a common Dance to indicate an intended 'diagonal step'. The set of proposed actions $P$ can be regarded as a set of vertices in $lk(v)$ of the current state $v$. If $P$ spans a simplex in $lk(v)$, then all actions can be safely performed simultaneously in the execution phase. If not, then we choose a maximal simplex in the subcomplex of $lk(v)$ spanned by $P$, yielding a maximal subset of simultaneously valid actions to be executed. More formally, we could regard such a 'collision avoidance policy' as a function from the power set of $V(lk(v))$ to $lk(v)$. Thus, we can reframe the problem of collision avoidance in terms of the combinatorial structure of links which, in turn, is captured by the local geometry of the state complex.

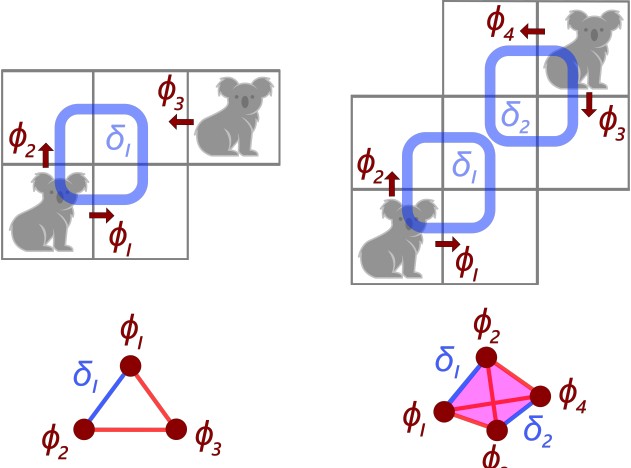

Figure 8: The two situations which lead to failure of Gromov's Link Condition in multi-agent gridworlds. Maroon arrows indicate admissible moves and blue squares indicate admissible dances. Note that in the links (bottom row), the triangle is missing in the left example, while the (solid) tetrahedron is missing in the right (however, all 2D faces are present). This is due to the respective collections of moves and dances failing to commute – an agent interrupts the other's dance (left) or two dances collide (right).

We now characterise the situations leading to failures of Gromov's Link Condition in the agent-only case. Before doing so, let us first classify low-dimensional simplices in $lk(v)$ for a vertex $v$ in our modified state complex $\mathcal{S}'$. A 0–simplex in $lk(v)$ corresponds to an admissible move at $v$. However, a 1–simplex either represents a pair of commuting moves, or two moves in a common dance. A 2–simplex either represents three agents moving pairwise independently, or a dancing agent commuting with a moving agent. Finally, a 3–simplex represents either four agents moving pairwise independently, one dancing agent and two moving agents that pairwise commute, or a pair of commuting dancers.

**Theorem 5.2** (Gromov's Link Condition in the modified state complex)**.** *Let $v$ be a vertex in the modified state complex $\mathcal{S}'$ of an agent-only gridworld. Then*

- *$lk(v)$ satisfies Gromov's Link Condition if and only if it has no empty 2–simplices nor 3–simplices, and*

- *if $lk(v)$ fails Gromov's Link Condition then there exist a pair of agents whose positions differ by either a knight move or a 2–step bishop move (as in Figure 8).*

*Proof.* If $lk(v)$ satisfies Gromov's Link Condition, then it has no empty simplices of any dimension, giving the forward implication. For the converse, assume that $lk(v)$ has no empty 2–simplices nor 3–simplices.

Suppose there exist $n + 1$ vertices spanning a complete subgraph of $lk(v)$, where $n \geq 4$. We want to show that these vertices span an $n$–simplex. By induction, we may assume that every subset of $n$ vertices from this set spans an $(n-1)$–simplex. Since $n \geq 4$, every quartuple of vertices in this subgraph spans a 3–simplex. Therefore, appealing to our classification of low-dimensional simplices, every pair of moves or dances involved has disjoint supports. Thus, the desired $n$–simplex exists. Consequently, potential failures can only be caused by empty 2–simplices or 3–simplices.

Next, we want to determine when three pairwise adjacent vertices in $lk(v)$ span a 2–simplex. These vertices represent three admissible moves at $v$. Since they are pairwise adjacent, they either correspond to three agents each doing a Move, or to one agent dancing with another one moving. In the former case, the supports are pairwise disjoint and so these moves form a commuting set of generators. Therefore, the desired 2–simplex exists (indeed, in the absence of dancers, the situation is the same as the original Abrams, Ghrist & Peterson setup). For the latter case, suppose that the first agent is dancing while the second moves. Since the 0–simplices are pairwise adjacent, each of the two admissible moves within the dance has disjoint support with the second agent's move. Thus, the only way the support of the dance fails to be disjoint from that of the second agent's move is if the second agent can move into the diagonally opposite corner of the dance. Therefore, the only way an empty 2–simplex can arise is if the agents' positions differ by a 'knight move' (see Figure 8 for illustration).

It remains to determine when four pairwise adjacent vertices in $lk(v)$ span a 3–simplex. We may assume that each triple of vertices in this set spans a 2–simplex, for otherwise we can reduce to the previous case. Let us analyse each case by the number of involved agents. If there are four involved agents, then each 0–simplex corresponds to exactly one agent moving. Since no dances are involved, it immediately follows that the desired 3–simplex exists. If there are three involved agents, then one is dancing while the other two move. Since each triple of 0–simplices spans a 2–simplex, we deduce that each move has disjoint support with the dance. Therefore, the dance and the two moves form a commuting set, and so the 3–simplex exists. Finally, if there are two agents then they must both be dancers. By the assumption on 2–simplices, each admissible move within the dance of one agent has disjoint support from the dance of the other agent. Thus, the only way for the two dances to have overlapping supports is if their respective diagonally opposite corners land on the same cell. Therefore, the only way an empty 3–simplex can arise (assuming no empty 2–simplices) is if two agents' positions differ by a '2–step bishop move' (see Figure 8 for illustration). $\qquad \square$

Consequently, if the Link Condition fails at all, it must fail at dimension 2 or 3. This can be interpreted as saying that we only need a bounded amount of foresight to detect potential collisions: under fog-of-war, each agent needs a line-of-sight of only four moves.

Positive curvature could indicate collisions between any specified labels (e.g., objects), however, for this interpretation to be valid we would need to carefully identify which other potential cycles in the state complex ought to be filled in. Doing this in a 'natural' way is in itself a non-trivial task, and is the subject of further investigation.

## 6 Experiments and applications

Although our main contribution is theoretical, we conduct some small initial experiments to demonstrate the type of information which can be captured in the geometry and topology (see Appendix A.1). To run these experiments, we developed and used a custom Python-based tool (detailed in Appendix A.2). Our focus on small rooms is largely expository, i.e., they are the simplest non-trivial examples illustrating the key features we want to isolate, and naturally reoccur in all larger rooms. Our intention is also to demonstrate a combinatorial explosion in the number of states. We don't recommend constructing the entire state complex in practical applications (indeed, to implement addition of integers on a computer, it is infeasible and unnecessary to construct *all* integers).

*Remark* 6.1. By a simple counting argument, one can deduce the total number of states in a gridworld. For an agent-only gridworld with $n$ cells and $k$ agents, there is a total of $\binom{n}{k}$ states. If there are $n$ cells, $k$ agents, and $j$ objects, then there are $\binom{n}{k}\binom{n-k}{j}$ states. Thus, even for a moderately sized $10 \times 10$ room with 50 agents, there are $\binom{100}{50} \approx 1.008 \times 10^{29}$ vertices in the state complex.

By Theorem 5.2, checking if $lk(v)$ satisfies Gromov's Link Condition requires computing the link only up to dimension 3 and then checking whether it is a flag complex; if not, we count the number of empty simplices. Checking this for an individual vertex in the state complex is not too computationally demanding, however doing so across the entire state complex becomes more difficult due to the combinatorial explosion in the number of states as the number of agents or room size grows. In practical applications, such as calculating collision-avoiding navigation routes, it is – again, by Theorem 5.2 – only necessary to construct a small local subcomplex. But perhaps even more importantly, to detect potential collisions between agents, it is not even necessary to construct $lk(v)$, since Theorem 5.2 provides a computational shortcut: just check for supports of knight or two-step bishop moves between agents.

To quantify the reduction in the computation workload, let us estimate the potential number of simplices in the link vs. the number of those which actually require checking. The number of $k$-simplices in the link should be on the order of at most $n^{k+1}$, however, we don't need to check these for $k > 3$ thanks to Theorem 5.2. The number of potential 3–simplices in the link could be quartic in the number of agents (arising due to 4 agents moving); however, there are at most quadratically many that require checking since a failure can only arise due to a pair of agents, again by Theorem 5.2. Similarly, the number of potential 2–simplices is at most cubic, but only quadratically many of these need to be checked. Consequently, checking Gromov's Link Condition at a given state is at most quadratic in the number of agents (due to the number of possible pairs).

By using Gromov's Link Condition, we can identify a precise measure of how far ahead agents ought to look in order to safely proceed without fear of collisions. Table 1 gives a summary analysis of a $3 \times 3$ room with varying numbers of agents. We noticed several symmetries. Commuting moves and the number of states have a symmetry about 4.5 agents (due to the label-inversion symmetry as previously illustrated in Figure 5). However, curiously, the number of dances has a symmetry about 3.5 agents. This difference leads to the asymmetrical distribution of positive curvature and failures of Gromov's Link Condition – which, while maximal for 3 agents as a proportion of total states, exhibited the highest mean failure rate for 4 agents.

This shows that, heuristically, we expect most states to satisfy NPC (see Table 1), and so existing greedy algorithms (Ardila et al., 2012) for calculating geodesics will work well in most situations. However, to implement an efficient, collision-free path-finding algorithm in our modified state complexes, we need to add an additional check. Specifically, when we are near a potentially dangerous state, we should implement a predefined 'detour' to avoid the collision, which can be done on a local basis using the identified supports which lead to positive curvature (as in Figure 8).

Table 1: Data of Gromov's Link Condition failure and commuting 4–cycles in the state complexes of a $3 \times 3$ room with varying numbers of agents and no objects. The percentage of NPC states (shown in brackets in the second column) is rounded to the nearest integer. The mean number of Gromov's Link Condition failures (shown in the penultimate column) is the mean number of failures over the total number of states, and is rounded to two decimal places.

| Agents | States (% NPC) | Dances | Commuting moves | Gromov's Link Condition Failures | | |
| --- | --- | --- | --- | --- | --- | --- |
| | | | | Total | Mean | Max |
| 0 | 1 (100) | 0 | 0 | 0 | 0 | 0 |
| 1 | 9 (100) | 4 | 0 | 0 | 0 | 0 |
| 2 | 36 (78) | 20 | 44 | 32 | 0.89 | 4 |
| 3 | 84 (62) | 40 | 220 | 184 | 2.19 | 14 |
| 4 | 126 (65) | 40 | 440 | 288 | 2.29 | 11 |
| 5 | 126 (68) | 20 | 440 | 152 | 1.21 | 6 |
| 6 | 84 (86) | 4 | 220 | 16 | 0.19 | 2 |
| 7 | 36 (100) | 0 | 44 | 0 | 0 | 0 |
| 8 | 9 (100) | 0 | 0 | 0 | 0 | 0 |
| 9 | 1 (100) | 0 | 0 | 0 | 0 | 0 |

# 7 Conclusions and future directions

This study presents novel applications of tools from geometric group theory and combinatorics to the AI research community, opening new ways for recasting and analysing AI problems as geometric ones. Using these tools, we show an example of how the intrinsic geometry of a task space serendipitously embeds safety information and makes it possible to determine how far ahead in time an AI system needs to observe to be guaranteed of avoiding dangerous actions.

Leike et al. (2017) show deep reinforcement learning agents cannot solve many AI safety problems specified on gridworlds, e.g., minimising unwanted side-effects or ensuring robustness to agent self-modification. Having described the agent-only case in this study, there is now ripe opportunity to account for positive curvature or other geometric features arising due to other labels or generators (actions) present in specified AI safety problems, e.g., agents pushing/pulling objects, pressing buttons, modifying their form or behaviour, rewards/punishments, opening/unlocking doors, etc.. By considering *directed* modified state complexes, irreversible actions can be captured by 'invariant subcomplexes' (i.e., you can't escape from them), allowing geometric study of the tree/flowchart of irreversible actions and related recurrence/transience. Braiding can be used to study route planning, back-tracking, cooperation, assembly, and topological entropy in congestion (Ghrist, 2009). Numerous extensions are possible, allowing us to study and geometrically represent further problems with a view to developing efficient, geometrically-inspired local algorithms without the need for training.

Do learning algorithms already implement such geometrically-inspired algorithms, the related geometry, or approximations thereof? To find out, it will be interesting to investigate how modified state complexes map to learned internal representations of neural networks trained to predict multi-agent gridworld dynamics. Such mappings would connect the geometry and topology of a task space directly to optimisation procedures and learning trajectories in latent representation spaces, potentially highlighting topological and geometric differences and opportunities for deeper insight and improvement of optimisation procedures, in the spirit of Naitzat et al. (2020); Zhao & Zhang (2022). One could also compare biological optimisation processes and internal representations of allocentric and egocentric navigation (Burgess, 2006; Gardner et al., 2022), and how this interacts with the position of other agents (Duvelle & Jeffery, 2018; Sutherland & Bilkey, 2020). Additionally, geometrically integrating safety information to constrain learning and control algorithms (Verginis et al., 2021) may be fruitful.

From a more mathematical perspective, state complexes of gridworlds give rise to an interesting class of geometric spaces. It would be worthwhile to investigate their geometric and topological properties to more deeply understand various aspects of multi-agent gridworlds. For example, for a gridworld with $n$ agents in a sufficiently large room, we hypothesise that the modified state complex should be a classifying space for the $n$–strand braid group. This is clearly false when the room is packed full of agents (in which case the state complex is a single point), so it may be fruitful to determine if there is some 'critical' density at which a topological transition occurs.

Using the *failure* of Gromov's Link Condition in an essential way appears to be a relatively unexplored approach. Indeed, much of the mathematical literature concerning cube complexes focusses on showing that the Link Condition always holds. In works such as Abrams & Ghrist (2004); Ghrist & Peterson (2007); Ardila et al. (2012) NPC is a *property* which may be exploited using subsequent related properties of the cube complex (e.g., known hyperplane arrangements or CAT(0) geometry) to prove statements about the modelled systems. In contrast, since NPC does not always hold in our modified state complexes, we can instead use it (or its failure) as a *condition* to detect some interesting features; namely, potential collisions of agents. To our knowledge, the only other works which go against this trend are Abrams & Ghrist (2004), in which failure detects global disconnection of a metamorphic system, and Bell et al. (2019), where failure detects non-trivial loops on topological surfaces. It would be interesting to explore cube complexes arising in other settings where failure captures critical information.

Another interesting mathematical direction to explore is the potential connection of (modified) state complexes and the work of Ghrist & Lavalle (2006). In Ghrist & Lavalle (2006), a 'roadmap co-ordination space' is constructed by taking a product of $n$ graphs, then deleting an 'obstacle' set. Here, each graph involved

represents a 'roadmap' where the corresponding agent can move. One may adapt this to the setting of $n$ (distinguished) agents in a gridworld: choose the gridworld as the roadmap for each agent, then delete cubes when there is a collision between two agents (this needs to be done carefully). We expect the resulting roadmap co-ordination space to be the same as the state complex for the (labelled) $n$ agent gridworld obtained by directly constructing the (unmodified) state complex. In other words, we believe the Ghrist–Lavalle and Abrams–Ghrist–Peterson approaches should both yield the same object in the context of distinguished agents in a gridworld. Under either approach, one obtains an object which is NPC. However, both frameworks suffer from a common limitation: each agent is confined to move on a graph – a 1–dimensional space. However, as we argue in Section 4, a gridworld ought to be regarded as a discretised 2–dimensional domain. In this light, it would be interesting to define a 'modified roadmap configuration space' which allows each roadmap to be a (discretised) rectangle (or whatever shape the room might be). It seems plausible that by deleting 'collision' cubes in an appropriate manner, one should recover our modified state complex (for $n$ labelled agents). If so, it would further support the case that filling squares whenever there are dances is the most natural approach. Indeed, our guiding intuition for the modified state complex of an $n$-agent gridworld is that it should serve as a discrete analogue of the configuration space of $n$ points on the plane.

### Broader Impact Statement

A limitation of our work is that we have so far only explored very simple AI environments. Further work is needed to expand the framework and results to more general, sophisticated, and real-world environments. For this reason, although our work provides new geometric perspectives, data, and potential algorithms for an important AI safety issue, we caution against hasty real-world implementation of the main results. To avoid potential negative societal impacts, it would still be important to perform rigorous checks and tests in application domains, since our results do not directly extend to situations beyond which the stated assumptions hold.

### Acknowledgements

This project began as a rotation project during the first author's PhD programme. We wish to acknowledge and thank Anastasiia Tsvietkova for supporting the first author to do a research rotation in the Topology and Geometry of Manifolds Unit at OIST, in which the second author was a postdoctoral scholar. The first author thanks Nick Owad for 3D printing a model state complex. The second author acknowledges the support of the National Natural Science Foundation of China (NSFC 12101503) and the Suzhou Science and Technology Programme (ZXL2022473). We thank anonymous reviewers for their suggestions to improve exposition. The authors declare no other competing interests.

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

# A   Appendix

## A.1   Gallery.

The $2 \times 3$ room with two agents shows multiple instances of local positive curvature in the associated modified state complex. Figure 9 shows one such state where Gromov's Link Condition fails due to the agents being separated by a knight's move (see Theorem 5.2). At this state, there are actually two empty 2–simplices in its link – this is because the pattern appearing in the 5–cell subgrid with two agents (as in Figure 8) arises in two different ways within the given state on the gridworld. The only other state where Gromov's Link Condition fails is a mirror image of the one shown.

Further small gridworlds and their respective state complexes are shown in Figures 10, 11, and 12.

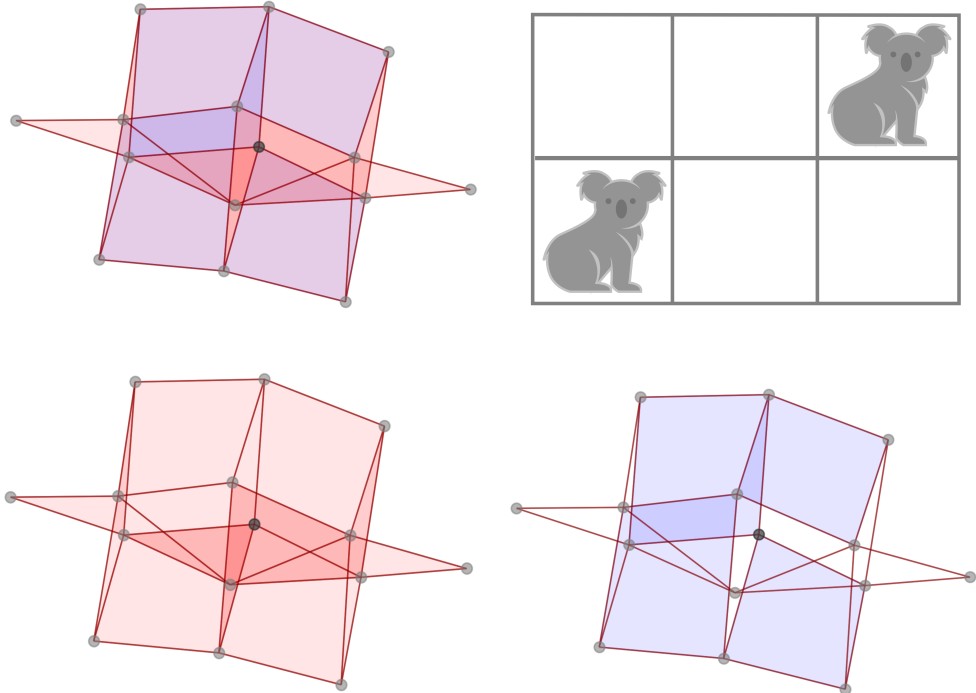

Figure 9: A $2 \times 3$ room with two agents (top right) and its state complex (top left), where dances are shaded blue and commuting moves are shaded red. The darker-shaded vertex represents the state of the gridworld shown. Also shown is the state complex with only commuting moves (bottom left) and only dances (bottom right).

## A.2   Python tool for constructing gridworlds and their state complexes

We developed a Python-based tool for constructing gridworlds with objects and agents. It includes a GUI application for the easy specification of gridworlds and a script which will produce plots and data of the resulting state complex. We ran all experiments on a Lenovo IdeaPad 510-15ISK laptop. The open-source code is available here: https://github.com/tfburns/State-Complexes-of-Gridworlds.

For the sake of generality and future-proofing of our software, we chose to construct the links in our implementation of checking Gromov's Link Condition in gridworlds, which is not necessary in-practice. Instead, in practical situations, one can directly check for supports of knight or two-step bishop moves between agents, which per Theorem 5.2 provides a computational short-cut for detecting failures in agent-only gridworlds. Another area of computational efficiency available in many rooms are in the symmetries of the

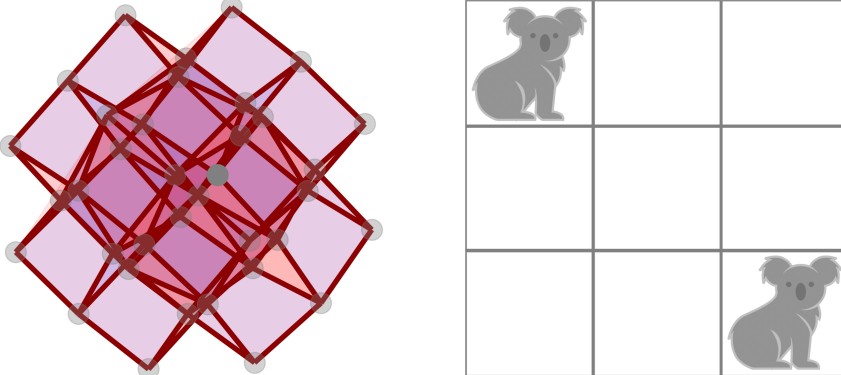

Figure 10: A $3 \times 3$ room with two agents (right) and its state complex (left), where dances are shaded blue and commuting moves are shaded red. The darker-shaded vertex represents the state of the gridworld shown. Naturally-occurring copies of this state complex can be found as sub-complexes in the state complex shown in Figure 12.

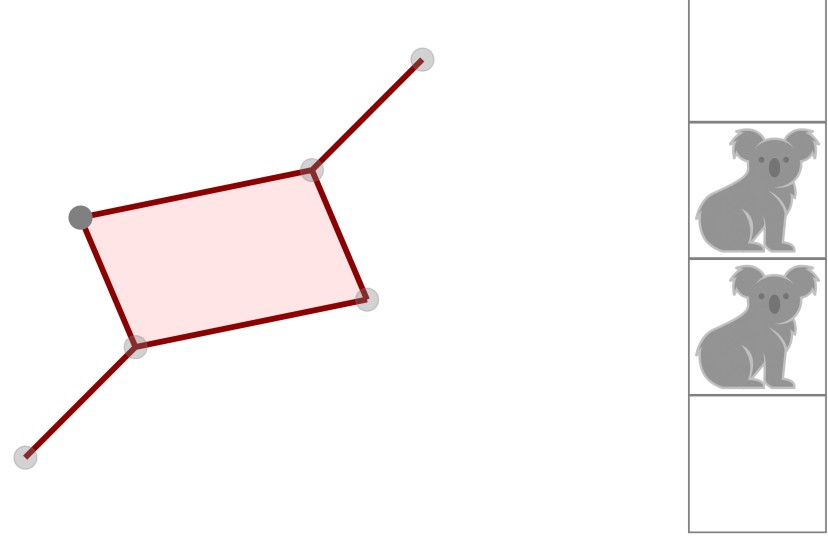

Figure 11: A $4 \times 1$ corridor with two agents (right) and its state complex (left). There are no dances and only one commuting move, shaded red. The darker-shaded vertex represents the state of the gridworld shown. Naturally-occurring copies of this state complex can be found as sub-complexes in the state complex shown in Figure 12.

room itself. For example, an evenly-sized square room can be cut into eighths (like a square pizza), where each eighth is geometrically identical to every other.

Users of the code will notice a small but important implementation detail in the code which we chose to omit the particulars of in this paper: in the code, we need to include labelled walls along the borders of our gridworlds. This is because we construct our gridworlds computationally as coordinate-free, abstract graphs. For Move, the lack of a coordinate system is not an issue – if an agent label sees a neighbouring vertex with an empty floor label, the support exists and the generator can be used. However, Push/Pull only allows objects to be pushed or pulled by the agent in a straight line within the gridworld. We ensure

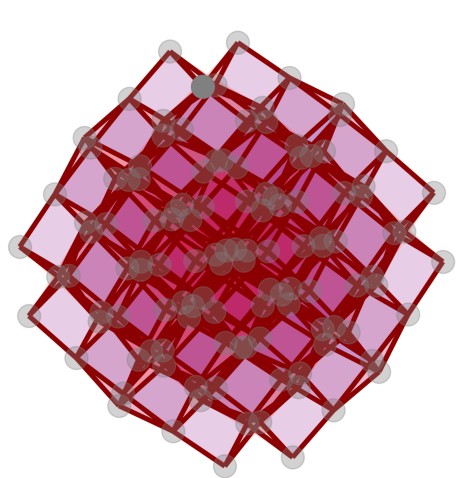 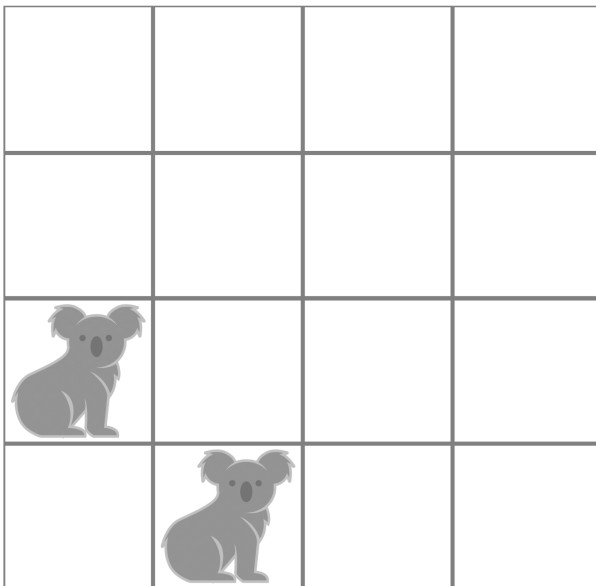

Figure 12: A $4 \times 4$ room with two agents (right) and its state complex (left), where dances are shaded blue and commuting moves are shaded red. The darker-shaded vertex represents the state of the gridworld shown. Embedded within this state complex are naturally-occurring copies of the state complex of the $4 \times 1$ corridor with two agents, shown in Figure 11. There are also naturally-occurring copies of state complex of the $3 \times 3$ room with two agents, shown in Figure 10.

this straightness in the abstract graph by identifying a larger subgraph around the object and agent than is illustrated in Figure 2. Essentially, we incorporate three wildcard cells (cells of any labelling) adjacent to three labelled cells ('agent', 'object', and 'floor'), such that together they form a $2 \times 3$ grid.

### A.3 Classification of $4$–cycles

A $4$–cycle in the state graph could arise in several different ways. For example, it could arise from a pair of commuting moves, one agent dancing, or two agents doing a half-revolution about one another as in Figure 3. To distinguish these in our implementation for agent-only gridworlds, we count the number of times a cell is occupied by an agent within the $4$–cycle. We use this fact in our Python tool to identify commuting move and dancing $4$–cycles.

For the remainder of this section, let $v_0, v_1, v_2, v_3$ denote the vertices of an embedded $4$–cycle in the modified state complex of an agent-only gridworld, and suppose $\phi_i$ is the generator such that $\phi_i[v_i] = v_{i+1}$ (modulo 4) for each $i$. Let $U := \cup_i SUP(\phi_i)$.

**Lemma A.1** (Supports of $4$–cycles). *There are exactly 4 cells contained in $U$.*

*Proof.* Consider applying the generators $\phi_0, \phi_1, \phi_2, \phi_3$ sequentially starting from the initial state $v_0$. Each $\phi_i$ is a Move generator which swaps the 'agent' and 'floor' labels in a pair of adjacent cells. Since we return to $v_0$ at the end of this $4$–cycle, each cell in $U$ appears in an even number of the $\phi_i$'s. Therefore, $U$ comprises at most 4 cells.

Suppose for a contradiction that $U$ contains exactly $n \leq 3$ cells. Suppose that there are $k$ 'agent' labels appearing on any (hence every) restricted state $v_i|_U$. Then there are $\binom{n}{k}$ possible ways to label the cells of $U$ with exactly $k$ agents. Since $\binom{n}{k} \leq 3$ for all $n \leq 3$ and $0 \leq k \leq n$, it follows that the $v_i|_U$ cannot all be distinct. Since all $v_i$ agree outside $U$, we deduce that $v_0, v_1, v_2, v_3$ cannot all be distinct. This contradicts the assumption that they form an embedded $4$–cycle. $\square$

**Lemma A.2** (4–cycle classification)**.** *Any embedded* 4*–cycle in the modified state complex of an agent-only gridworld arises from either:*

- *a single agent doing a dance,*

- *a pair of commuting moves,*

- *a pair of agents doing a half-twist in a* $2 \times 2$ *subgrid, or*

- *three agents doing an 'inverted' dance.*

*Proof.* We consider the cases depending on the number of involved agents, that is, the number of '`agent`' labels appearing on $v_0|_U$.

If there is only one involved agent, then this agent must occupy each cell in $U$ exactly once throughout the 4–cycle. Since the agent must Move to an adjacent cell at each step, it must move along an embedded 4–cycle in the (dual graph of the) gridworld. The only way this can occur is if $U$ is a $2 \times 2$ subgrid. Therefore, the agents performs a dance.

The case of three involved agents follows analogously by inverting the '`agent`' and '`floor`' labels.

Now suppose there are two agents involved. After performing the 4–cycle, the two agents either return to their starting locations, or swap their positions. In the former case, each agent must move exactly twice throughout the 4–cycle. The only way this can occur is if each agent moves to an adjacent cell and then moves back to its initial position. Since $U$ comprises exactly 4 cells, the supports of the respective Moves must be disjoint. Therefore, the agents perform a pair of commuting Moves.

It remains to deal with the case where the agents swap positions. Let us view $U$ as an induced subgraph in the gridworld. Since the agents swap positions, $U$ must be connected. Since $U$ has 4 cells, it must be isomorphic to either a path graph of length 3, a 'tripod' (a tree with one vertex of valence 3 and three vertices of valence 1), or a cycle graph of length 4. If $U$ is a path graph then it is impossible for the agents to swap positions. The state graph for two agents in a tripod is isomorphic to a cycle graph of length 6 (as in Figure 13), therefore the agents cannot swap positions using only 4 Moves. It follows that $U$ must be a cycle graph, and so the involved cells form a $2 \times 2$ subgrid. Therefore, our problem reduces to examining 4–cycles in the state complex for two agents in a $2 \times 2$ room, as depicted in Figure 3. The 4–cycles where the agents swap positions are precisely those that are not filled in with a square. In each of these cases, our 4–cycle arises from two agents performing a 'half-twist' in a $2 \times 2$ subgrid. □

**Lemma A.3** (4–cycle identification)**.** *The type of an embedded* 4*–cycle in the modified state complex of an agent-only gridworld can be identified by counting the occurrences each involved grid cell is occupied by an agent. Specifically, we have*

- *a dance* $\iff$ *one count in each of the four cells,*

- *commuting moves* $\iff$ *two counts in each of the four cells,*

- *a half-twist* $\iff$ *three counts in one cell, two counts in two cells, one count in one cell, and*

- *an inverted dance* $\iff$ *three counts in each of the four cells.*

*Proof.* This follows by inspecting each case as described in Lemma A.2 and counting the number of occurrences each grid cell is occupied by an agent. □

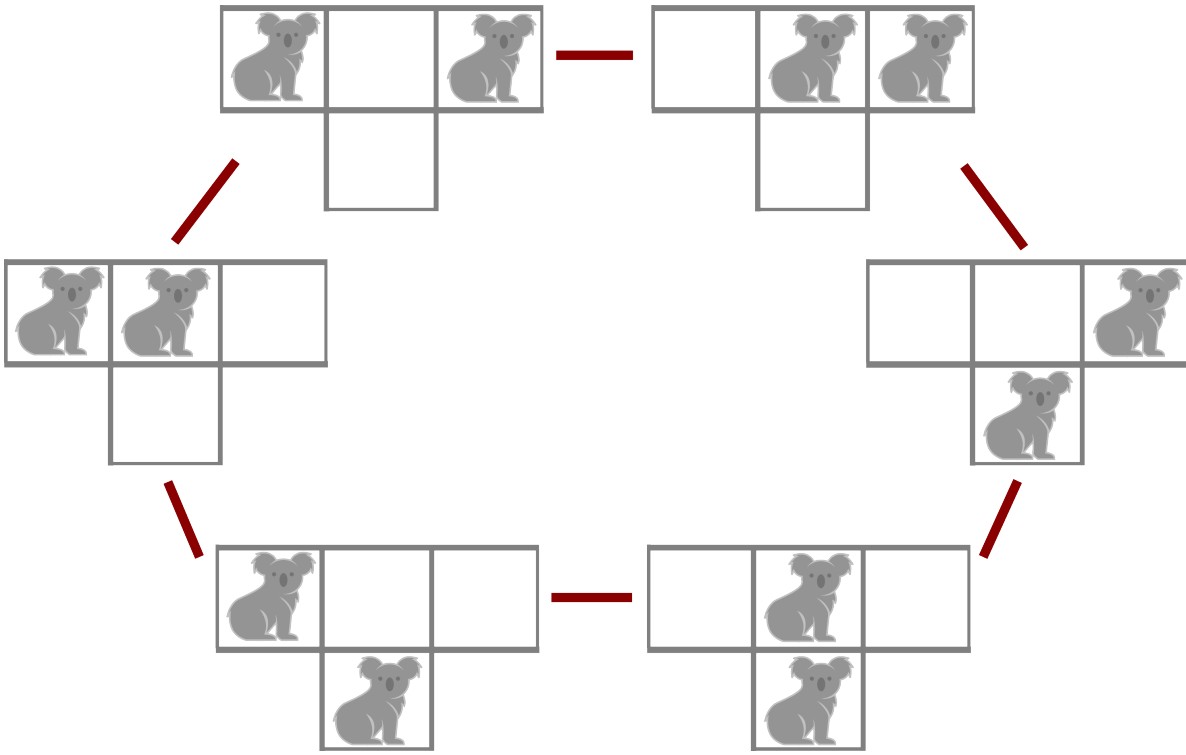

Figure 13: The state complex of a T–shaped room with two agents, forming a 6–cycle.

