# OpenReview forum: "Detecting danger in gridworlds using Gromov’s Link Condition"
_TMLR — Accepted by TMLR_

### Review · Reviewer_2GJX · 2023-09-04

**Summary Of Contributions:**

The paper adopts the framework of reconfigurable systems and state complexes to the study of gridworlds. Using a collection of figures, the paper offers a friendly introduction to the construction of state complexes for simple environments with agents and movable objects, with additional visualizations of the resulting state ~graphs~ complexes.

The main result is a theoretical characterization of local failures of the Gromov link condition, and the implied non-positive curvature (NPC), for a modified state ~graph~ complex. The modification is proposed to remove uninteresting topology of 4-cycles that don't bound an actual obstacle or void, where agents are only allowed to move horizontally or vertically to empty cells within the grid. Since NPC served as the basis of multiple key results in reconfigurable systems, this deviation is potentially interesting. (Note that the authors only derive this characterization for agent-only environments.)

The paper concludes with statistics of NPC failures in a fixed grid with varying number of agents.

**Audience:**

Yes

**Broader Impact Concerns:**

Adequately addressed by the authors.

**Claims And Evidence:**

No

**Requested Changes:**

**Claims:**
- [abstract] *More broadly, our work introduces tools from geometric group theory and combinatorics to the AI community*
  - For this claim to hold, I believe the paper needs to make a better job rigorously presenting background mathematical tools with a more thorough survey of how those tools were applied to related problems, e.g., in motion planning. See comments below for presentation issues.
- [page 1] *Our main contribution is the first application of this geometric approach (of using state complexes) to the setting of multi-agent gridworlds.*
  - For this claim to hold, the paper needs to make concrete progress on a fundamental problem in the study of gridworlds using the geometric approach in question. In contrast, experimental results only report statistics for newly-introduced notions. In addition, state complexes have been utilized numerous times in the study of multi-agents in very similar settings.
- [page 1] *Nevertheless, by applying Gromov’s Link Condition, we completely characterise when positive curvature occurs in our new state complexes*
  - This claim is overstated (completely). This result was only derived for an agent-only environment, which does not qualify as a complete characterization.
- [page 2] *Current approaches to collision-detection and navigation during multi-agent navigation often rely on modelling and predicting collisions based on large training datasets [..] or by explicitly modelling physical movements [..]*
  - This claim is overstated (often). Prior work applied other theories, including the ~NPC~ Gromov link condition~s~ studied in this paper.
- [page 2] *Our work relates to a growing body of research aimed towards understanding, from a geometric perspective, how deep learning methods transform input data into decisions, memories, or actions*
  - This framing is difficult to justify given the actual content of the article. I would recommend deferring this connection to the discussion or future work sections at the end.
- [page 2] *However, such studies do not usually incorporate the geometry of the originating domain or task in a substantial way, before applying or investigating the performance of learning algorithms – and even fewer do so for multi-agent systems. One possible reason for this is a lack of known suitable tools.*
  - This claim lacks evidence. There has certainly been a lot of work in machine learning on understanding the geometry of the input data and how it impacts learning algorithms, e.g., manifold hypothesis.

**Missing technical discussion:**
- Section 2
  - It is necessary to explain explicitly how generators stand for the more common notion of turn-taking. That is, the state complex also discretizes time from the perspective of agents.
  - Also explain how the notion of commutativity relates to simultaneous action by independent agents, i.e., agents perceiving the system at a given state can decide to move simultaneously if the resulting actions commute.
  - This leaves non-commuting moves under-specified. From a physical viewpoint, those aren't realizable without collision. It is not clear if the resulting (invalid) states are included in the state graph, for otherwise the simultaneous execution of a non-commuting set of moves cannot happen by definition. For example, fewer legal actions can be selectively allowed, e.g., by tie-breaking, which seems to be how many computer games resolve this issue.
  - The above setting is both natural and practical, and similar formulations were studied in prior works, for example:
    - Ghrist, Robert, and Steven M. Lavalle. "Nonpositive curvature and pareto optimal coordination of robots." SIAM Journal on Control and Optimization 45, no. 5 (2006): 1697-1713.
    - Please cite and discuss, clearly positioning the proposed contributions in this light.
  - Perhaps discussing 'fog of war' at this point would help clarify the scope of the contribution.
- Section 4
  - Please explain explicitly whether dances allow agents to move diagonally. If so, wouldn't it be more natural to include diagonal moves from the get-go?
  - It appears dances only allow diagonal moves if the move can be executed both as a sequence of horizontal+vertical or vertical+horizontal moves, i.e., if only one of those two sequences are admissible, the diagonal move cannot happen. I don't see a justification for this restriction, since the stated purpose of dances is to more faithfully describe the continuum of a 2D plane -- following the same reasoning at the bottom of Page 6.
- Section 5
  - Please note that the article makes no mention of CAT(0).
  - Would it help to use computer-aided proofs or enumeration to characterize NPC failures involving objects or multiple agents+objects?

**Presentation Issues:**
- The notation $\mathcal{S}^{(1)}$ was not justified.
- The discussion (3 paragraphs) near Figure 3 seems too verbose, and seems to reappear each time a complex is defined.
  - I would recommend presenting a formal definition as a simple equation.
- In justifying small environments (first paragraph of Section 6) I found the analogy to integer addition unnecessary. The remark was much more relevant to the discussion.
- It is not immediately clear how label-inversion and loss of symmetry help the main development, or fit the same motivations towards practical applications.
  - I would recommend deferring to an appendix with more space to say more for motivation, or at least limiting such discussion to specific points where they are deemed most relevant.
- Section 3 appears largely unrelated to the development towards the main result, and does not offer specific technical contributions.
  - Breadth-first-search is a standard technique, and limiting the study to connected state graphs makes this less interesting.
  - It may be true that visualizations can be revealing for small subsets of the state ~graph~ complex, but this point was not adequately developed.
  - The section seems to report a collection of observations before digressing into further unanswered questions.
  - I would strongly recommend to defer this discussion and related figure to an appendix.
- Last paragraph of Section 4 seems to digress beyond the scope of the paper: ``*This can be used to study the braiding or mixing of agents, and also allows us to consider path-homotopic paths as ‘essentially’ the same.*''

**Strengths And Weaknesses:**

*Positives:*
- Adopts a principled approach to safe planning in gridworlds, based on the mathematical framework of reconfigurable systems.
- Identifies a potentially interesting connection between NPC failures and undesirable states.
- Mentions connections to the notion of 'fog-of-war' from real-time strategy games.
- Proposes simple local tests to detect/anticipate undesirable states, e.g., in an AI planner.

*Needs work:*
- Key definitions need to be clarified; see below.
- ~The main result, which can be seen in Figure 8, is fairly obvious, which sheds doubts on the potential of the proposed approach to bring about fundamental progress in this application area. This is not helped by the fact that the state complex is prohibitively large, even when restricted to small neighborhoods or scenarios, which hinders its adoption in practice.~
- The main result, as summarized in Figure 8, appears obvious. This leaves doubts as to the how much it helps in practice. It would help greatly to demonstrate concrete benefits in an AI planning application, beyond the statistics reported in the experimental section.
- Positioning with respect to prior work needs to be clarified as well; see below.

---

> ### Author Response · Authors · 2023-09-25
> **Response 1 to Reviewer 2GJX**
>
> We thank Reviewer 2GJX for their thorough review. Below we provide general responses to some larger points and themes. Below that, we provide specific responses to some particular points raised. Given Reviewer 2GJX suggests and requests several changes, we wish to use the discussion period to clearly identify and describe the proposed changes to improve the manuscript. In this spirit, any and all content in these responses can be considered for inclusion into the text, along with specific changes and inclusions which are mentioned.
>
> **General responses:**
>
> **State graphs vs. state complexes**
>
> Throughout the review, Reviewer 2GJX refers to our work on ‘state graphs’ and appears to use ‘state graphs’ and ‘state complexes’ interchangeably. It is important to note that these are different but related objects. In particular, a state graph is the $1$-skeleton of a state complex. We do not, for example, introduce a ‘modified state graph’ (as Reviewer 2GJX stated in their ‘Summary Of Contributions’), but rather a modified state complex. Some of the requested changes and critiques seem to stem from this conflation, e.g., Reviewer 2GJX says that `state complexes have been utilized numerous times in the study of multi-agents in very similar settings`, however we are only aware of Ghrist and Lavelle [GL06] (which Reviewer 2GJX mentions in their review and that uses a similar but different setting). We ask the reviewer to please inform us of any other studies utilizing state complexes in such systems which they are aware of, as we would very much like to include them and this will improve the manuscript.
>
> **NPC as a condition vs. property**
>
> `[p]rior work applied other theories, including the NPC conditions studied in this paper`.
>
> Reviewer 2GJX mentions a related paper by Ghrist and Lavelle [GL06], which we will cite and discuss. However, as in other works we cited in the manuscript – Abrams and Ghrist (2004), Ghrist & Peterson (2007), and Ghrist (2009) – NPC is shown to be a *property*, not a condition. In using NPC as a property, these prior works exploit the subsequent related properties of the cube complex (e.g., known hyperplane arrangements or CAT(0) geometry) to prove statements about the modelled systems. In contrast, since NPC does not always hold in our modified state complexes, we can instead use it (or its failure) as a *condition* to detect some interesting features; namely, potential collisions of agents.
>
> We are unaware of prior studies which also use NPC (or its failure) as a condition in other applications. We would indeed be delighted to know about such work, as it would yield more interesting examples where some features in a system can be reformulated in geometric terms. We ask Reviewer 2GJX to please share with us the references to such work.
>
> **Additional mathematical background**
>
> Reviewer 2GJX states that `the paper needs to make a better job rigorously presenting background mathematical tools with a more thorough survey of how those tools were applied to related problems, e.g., in motion planning`. This seems at odds with another remark by the same reviewer, which was that `the paper offers a friendly introduction to the construction of state complexes for simple environments with agents and movable objects`. It also seems to contrast with positive comments from Reviewer jxVD, who said the paper has a `well-written introduction to this area`, and Reviewer oPeG, who stated as one of the strengths of the paper that `the writing is clear and concise, and translates complex math in a way that could be accessible for someone without as much formal training in that area`. Therefore, we ask Reviewer 2GJX to please clarify: what tools which were applied to, e.g., motion planning, were not sufficiently presented in terms of their mathematical background?

---

> > ### Comment · Reviewer_2GJX · 2023-09-26
> > **Final response to part-1 of the rebuttal**
> >
> > This is my final position on this part of the discussion. It is up to the authors to take my recommendations into account, in coordination with the action editor.
> >
> > **State graphs vs. state complexes**
> >
> > Thanks for pointing this out. I did largely speak about the two constructions interchangeably in my review, 3/5 times, but I'm not fully confused here. Now that the authors highlight the importance of this distinction, in regards to positioning wrt prior work, I request that the authors address this point explicitly in the introduction.
> >
> > **(Recommendation)** Please cite and briefly discuss related works in motion planning and AI that used state graphs, and make it clear how the submission and related works differ in utilizing state complexes.
> >
> > **NPC as a condition vs. property**
> >
> > No comment on this point of semantics. I edited my review to say `Gromove link condition` instead of `NPC conditions`.
> >
> > **Additional mathematical background**
> >
> > I described the presentation as friendly to be nice to the authors. This is not at odds as recommending a more thorough and rigorous account of preliminary results and further context from prior work, which I believe is needed to qualify for what the first claim promises.
> >
> > Interested readers, like reviewer jxVD would *need to* look up other papers to better appreciate some of the key concepts applied in this paper. As I mentioned, the notion of CAT(0) wasn't even discussed while it's likely to come up upon looking up related works. Indeed, the authors did say things like `We give an informal definition here, and refer to their papers for the precise formulation`. In contrast rigorous articles, e.g., Ghrist and Lavelle [GL06], have entire sections/subsections for definitions, and even *history* to really help introduce new tools into an existing field; see their S1.3, S2, and S4.1.
> >
> > **(Recommendation)** Please make sure to include sufficient rigorous definitions for the readers to understand the results, and how they fit within related results from prior work, to satisfy the first claim I called out in my review.

---

> > > ### Author Response · Authors · 2023-09-27
> > > **Follow-up response to ‘Final response to part-1 of the rebuttal’**
> > >
> > > Given Reviewer 2GJX states this was their `final position on this part of the discussion`, this response is mostly for the benefit of others to clarify misunderstandings and lay out our intended changes for the manuscript.
> > >
> > > **State graphs vs. state complexes**
> > >
> > > Although this distinction is already explicit in Section 1 (e.g., paragraph 2 starts with the sentence `In discrete settings, state spaces are typically represented by graphs or their higher dimensional analogues such as simplicial complexes or cube complexes.` and thereafter examples of `complexes` are discussed) and defined mathematically in Section 2 (Definitions 2.2 vs. 2.3), we are happy to follow Reviewer 2GJX’s recommendation to make this more explicit and `cite and briefly discuss related works in motion planning and AI that used state graphs`.
> > >
> > > **NPC as a condition vs. property**
> > >
> > > This is not a semantic point but a technical one. Reviewer 2GJX’s edited review now states that `Prior work applied other theories, including the ~~NPC~~ Gromov link conditions studied in this paper.` We asked Reviewer 2GJX to please share with us the references to such work, which they have not done. As stated in our original response: `We would indeed be delighted to know about such work, as it would yield more interesting examples where some features in a system can be reformulated in geometric terms.`
> > >
> > > **Additional mathematical background**
> > >
> > > Reviewer 2GJX asks for further mathematical background and reiterates that `the notion of CAT(0) wasn't even discussed while it's likely to come up upon looking up related works`. However, we already replied to this in our previous response (paragraph 1, Response 5):
> > >
> > > `For a cube complex X, the CAT(0) property is equivalent to X being NPC and simply connected (i.e. it has trivial topology). The AGP state complexes always satisfy NPC, but they are not necessarily simply connected. For example, the state complex for the 2x2 gridworld with 2 agents is homotopically equivalent to a circle, hence it is not simply connected, and therefore not CAT(0). Thus, we do not mention CAT(0) because it would be technically incorrect. We could mention that “NPC” is synonymous with “locally CAT(0)” for the benefit of readers who may have come across CAT(0) geometry. In the work of Ardila et al. on robotic arms, it is simple connectedness (a global property) which is harder to establish; the NPC property comes for free from AGP.`
> > >
> > > We therefore find this particular criticism regarding CAT(0) redundant, given Reviewer 2GJX did not respond to the above. And while we do reference some work for more technical details regarding AGP, we also provide many technical definitions and introductions throughout Sections 2, 4, and 5, and Appendix A3. We believe this balance is appropriate.

---

> ### Author Response · Authors · 2023-09-25
> **Response 2 to Reviewer 2GJX**
>
> **Specific responses:**
>
> **Weaknesses/needs work**
>
> `The main result, which can be seen in Figure 8, is fairly obvious, which sheds doubts on the potential of the proposed approach to bring about fundamental progress in this application.`
>
> This is yet another case of the all-too-familiar curse where a mathematical result turns out to be completely obvious in hindsight (especially given the proof). In actuality, we never suspected that the main result was true. Rather, our original goal was to prove that our modified state complexes were actually NPC. It was only in the process of failing to do so that we discovered the counter-examples illustrated in Figure 8. This was something we did not expect given that almost all “naturally-occuring” cube complexes in the mathematical literature are NPC. Only after some careful examination of these counter-examples did we realise the connection between NPC failure and collision-detection.
>
> Nevertheless, it is pleasing to see the result described as ‘obvious’ because this supports the naturalness of the result. Notably, we never sought out to capture safety information in the geometric representation, it just serendipitously ‘appeared’ as an embedded geometric property of the task space. The representation we were working towards was about finding a more faithful way to represent the topology of agent navigation in gridworlds. That this resulted in positive curvature at potential collision points to us only enhances the fittingness of the representation.
>
> We wish to remind Reviewer 2GJX of the TMLR acceptance criteria, which state that ‘novelty of the studied method is not a necessary criteria for acceptance’, and that ‘[i]f the authors make it clear that there is something to be learned by some researchers in their area from their work, then the criteria of interest is considered satisfied’. The criteria of interest seems satisfied given all three reviewers described clear strengths of the paper relevant to machine learning and AI, and which Reviewer 2GJX in particular described as:
> - `Adopts a principled approach to safe planning in gridworlds, based on the mathematical framework of reconfigurable systems.`
> - `Identifies a potentially interesting connection between NPC failures and undesirable states.`
> - `Mentions connections to the notion of 'fog-of-war' from real-time strategy games.`
> - `Proposes simple local tests to detect/anticipate undesirable states, e.g., in an AI planner.`
>
> As to whether or not our theoretical results have potential to `bring about fundamental progress in this application`, we point back to the strengths mentioned by Reviewer 2GJX and the other reviewers.
>
> `the state complex is prohibitively large, even when restricted to small neighborhoods or scenarios, which hinders its adoption in practice.`
>
> It is a common misconception that when working with systems with a very large, perhaps infinite, number of possibilities that computational approaches to solving problems would be intractable. If this were the case, then doing arithmetic on a computer would be considered impractical as there are infinitely many numbers, hence infinitely many possible sums to consider. We don't expect any practical implementation of addition or multiplication to involve generating the entire times-tables. Nor, in our case, do we recommend constructing the entire state complex in practical applications (as stated in paragraph 1, section 6). Instead, and by our theorem, we only need to compute a small neighbourhood of states when needed in practice (as stated in paragraph 3, section 6).
>
> Many examples where algorithms can be effectively implemented in situations involving infinite search spaces arise from decision problems in group theory. The Conjugacy Problem for a group $G$ asks: Given two elements $a,b \in G$, determine whether there exists an element $c \in G$ such that $ac = cb$. A famous result is that the Conjugacy Problem for braid groups can be solved in polynomial time [BKL98,FG03]. This does not involve naively searching the entire braid group for solutions (which is impossible since it is infinite). Instead, one exploits the underlying structure of the groups, using theoretical results to guarantee that if a solution exists, then it can be found using a bounded amount of work. In other words, you only need to generate enough data relevant to a given instance of the problem.

---

> > ### Comment · Reviewer_2GJX · 2023-09-26
> > **Final response to part-2 of the rebuttal**
> >
> > This is my final position on this part of the discussion. It is up to the authors to take my recommendations into account, in coordination with the action editor.
> >
> > ~Main result, per Figure 8, is obvious~
> >
> > **The main result, which can be seen in Figure 8, is fairly obvious** (actual quote from reviewer's original comment)
> >
> > Having a second agent within a given agent's 3x3 neighborhood easily anticipates collisions, and can be readily checked. No hindsight needed here. While a formal result based on established tools is appreciated, it is hoped that those results can be carried forward to novel situations of practical relevance. Similarly, while the author's personal experience developing the work is valuable, it's not necessarily the basis for e.g. motivating the work or evaluating when the result is mature enough for publication.
> >
> > **(Recommendation)** Please include justification for why this article qualifies as a first-step despite the fact that the results are still limited, both in terms of completeness (agent-only environment with two agents), and practicality (no AI planning results were presented, only statistics in a fixed grid world).
> >
> > My review makes no comments or recommendations regarding acceptance, and I'm very aware of the reviewing criteria for TMLR.
> >
> > As mentioned in my review, I still find analogies to integer addition unnecessary. Similarly, the Conjugacy Problem does not seem relevant here. The article resolved to only study the case of two agents in an obstacle free environment. Attempting to include more agents or obstacles is expected to quickly increase the combinatorial complexity of the search space. It is much more productive if the authors focus the discussion on those points, which are likely much more relevant to the readers. As mentioned in my review, the remark in Section 6 was indeed helpful in assessing the computational burden underlying the statistics presented.
> >
> > **(Recommendation)** Please include a statement about potential practical applications of similar formulations. Stating that only a small neighborhood needs to be explored is useful. It would help to estimate an upper bound on the number of states that needs to be explored, e.g., as a function of the neighborhood size and number of agents. It would be greatly helpful to mention whether this upper bound is polynomial or can be exponential.

---

> > > ### Author Response · Authors · 2023-09-27
> > > **Follow-up response 1 to ‘Final response to part-2 of the rebuttal’**
> > >
> > > Given Reviewer 2GJX states this was their `final position on this part of the discussion`, this response is mostly for the benefit of others to clarify misunderstandings and lay out our intended changes for the manuscript.
> > >
> > > **Main result, per Figure 8, is obvious**
> > >
> > > `Having a second agent within a given agent's 3x3 neighborhood easily anticipates collisions, and can be readily checked. No hindsight needed here.`
> > >
> > > We agree that any statement along the lines of “To detect potential collisions, check in a local neighbourhood for other agents” would be completely obvious. However, this is not the content of our main result (and was not what we were referring to when we mentioned hindsight), which we restate here:
> > >
> > > `Let $v$ be a vertex in the modified state complex $S’$ of an agent-only gridworld. Then
> > > - $lk(v)$ satisfies Gromov’s Link Condition if and only if it has no empty 2–simplices nor 3–simplices, and
> > > - if $lk(v)$ fails Gromov’s Link Condition then there exists a pair of agents whose positions differ by either a knight move or a 2–step bishop move (as in Figure 8).’
> > >
> > >
> > > `the results are still limited [...] in terms of completeness (agent-only environment with two agents)`
> > >
> > > This is a clear mischaracterisation of our work, and demonstrates a general misunderstanding of our contribution. Our Theorem applies to any finite number of agents; it can certainly be applied to environments which contain more than two agents (as shown in Table 1, our proof, and discussed throughout Sections 5 and 6). The point is that potential failures of Gromov’s Link Condition only requires us to check up to dimension 3 (note that the dimension of links could grow with the number of agents), and so we only need to check each pair of agents. Moreover, we prove that there can be no situations other than those in Figure 8 causing positive curvature. Thus, having another agent within a 3x3 neighbourhood does *not* lead to failure of Gromov’s Link Condition. We contend that our Theorem should not be technically obvious, especially the dimension 3 aspect.
> > >
> > > `the results are still limited [... in terms of …] practicality (no AI planning results were presented, only statistics in a fixed grid world)`
> > >
> > > Our Theorem provides a computational shortcut for checking Gromov’s Link Condition: just check up to dimension 3. This has clear practical applications, e.g., providing a bound on the computational requirements for calculating collision-avoiding navigation routes and identifying a precise measure of how far ahead agents ought to look in order to safely proceed without fear of collisions. Or, as Reviewer 2GJX notes, our work `Adopts a principled approach to safe planning in gridworlds` and `Proposes simple local tests to detect/anticipate undesirable states, e.g., in an AI planner`.

---

> > > ### Author Response · Authors · 2023-09-27
> > > **Follow-up response 2 to ‘Final response to part-2 of the rebuttal’**
> > >
> > > `As mentioned in my review, I still find analogies to integer addition unnecessary. Similarly, the Conjugacy Problem does not seem relevant here. The article resolved to only study the case of two agents in an obstacle free environment. Attempting to include more agents or obstacles is expected to quickly increase the combinatorial complexity of the search space. It is much more productive if the authors focus the discussion on those points, which are likely much more relevant to the readers.`
> > >
> > > We wish to clarify the repeated and extended mischaracterisations of our work: our Theorem applies for $n$ agents, and Sections 2 and 3 analyse gridworlds with both agents and objects. And it seems the computational discussion is still necessary (and should be expanded upon in the paper, as recommended by Reviewer 2GJX), given the persistent misconception that an excessively large search space would prohibit the implementation of practical algorithms. As we explained in our previous response regarding this point, `an intractably large (or infinite) system or state space, in itself, poses no inherent obstacle to implementing practical algorithms. What is crucial, however, are insights into the system, geometric or otherwise, as we provide in this case.` (paragraph 2, Response 3). Integer addition and solving the conjugacy problem in braid groups both have effective solutions despite the fact that they both take place on *infinite* search spaces. To give a more relevant example, Ardila et al. (2012) show that it is possible to calculate geodesics in arbitrarily large CAT(0) cubical complexes given certain insights and properties, where each iterative step is polynomial in time complexity.
> > >
> > > Likewise, by our Theorem, checking Gromov's Link Condition at a given state is at most quadratic in the number of agents (due to the number of possible pairs); this is *in spite* of the explosion in the combinatorial complexity of the search space with the number of agents. It is worth pointing out that the number of 3-simplices in the link could be quartic in the number of agents (arising due to 4 agents moving), but there are at most quadratically many that require checking, due to our Theorem. (The number of $k$-simplices in the link should be on the order of $n^{k+1}$, but we don’t need to check these for $k > 3$, again by our Theorem.) Thus, our theoretical results provide a significant reduction in the computation workload required to verify Gromov’s Link Condition. There was indeed a heavy computational burden when attempting to generate statistical information across the whole state complex (as the link of every vertex needs to be checked), which is why we don’t recommend constructing the entire search space in practical applications.

---

> ### Author Response · Authors · 2023-09-25
> **Response 3 to Reviewer 2GJX**
>
> This approach of computing only what is necessary, only when necessary, is a fundamental principle behind lazy evaluation. A key feature of lazy evaluation is its ability to handle potentially infinite data structures. This is exemplified by the topological software package Bigger [Bel21], which is capable of doing computations on surfaces of infinite topological type (i.e., having infinitely many handles or holes). Thus, even for problems or systems that are intrinsically infinite, it is still possible to implement them so long as one can find suitable data structures (though this requires some careful thought and insight).
>
> These examples demonstrate that an intractably large (or infinite) system or state space, in itself, poses no inherent obstacle to implementing practical algorithms. What is crucial, however, are insights into the system, geometric or otherwise, as we provide in this case.
>
> **Claims**
>
> `the paper needs to make concrete progress on a fundamental problem in the study of gridworlds using the geometric approach in question. In contrast, experimental results only report statistics for newly-introduced notions.`
>
> As the claim states ‘[o]ur main contribution is the first application of this geometric approach (of using state complexes) to the setting of multi-agent gridworlds’, which is true. No prior work that we are aware of has studied the setting of multi-agent gridworlds, nor with the geometric approach developed and described here, i.e., using NPC as a condition in our modified state complexes (or something akin to such complexes). If we are incorrect, we ask Reviewer 2GJX to kindly share with us references to such work that previously did this and we will gladly amend this claim and cite the work(s).
>
> ` 'Nevertheless, by applying Gromov’s Link Condition, we completely characterise when positive curvature occurs in our new state complexes'
> This claim is overstated (completely). This result was only derived for an agent-only environment, which does not qualify as a complete characterization.`
>
> When we stated this claim, we used ‘our new state complexes’ to refer to the modified state complexes for agent-only gridworlds. We agree that it is better to clarify this more explicitly, and propose to append ‘in the agent-only case’ to the originally-quoted sentence.
>
> ` 'Our work relates to a growing body of research aimed towards understanding, from a geometric perspective, how deep learning methods transform input data into decisions, memories, or actions'
> This framing is difficult to justify given the actual content of the article. I would recommend deferring this connection to the discussion or future work sections at the end.`
>
> It is unclear to us whether Reviewer 2GJX disagrees with this ‘claim’ or not and, if so, for which part(s) and for what reason(s). Reviewer jxVD stated this was `the most promising [direction] for further future work`, and we concur with the later comment by Reviewer 2GJX that `[t]here has certainly been a lot of work in machine learning on understanding the geometry of the input data and how it impacts learning algorithms, e.g., manifold hypothesis`. Therefore, and without clarity on what part of the ‘claim’ is objectionable, it seems appropriate to keep this sentence in the abstract to alert readers to this connection.
>
> ` 'However, such studies do not usually incorporate the geometry of the originating domain or task in a substantial way, before applying or investigating the performance of learning algorithms – and even fewer do so for multi-agent systems. One possible reason for this is a lack of known suitable tools.'
> This claim lacks evidence. There has certainly been a lot of work in machine learning on understanding the geometry of the input data and how it impacts learning algorithms, e.g., manifold hypothesis`
>
> It is true that much work has been done on the geometry of data and tasks, especially related to the neural representations thereof, e.g., the manifold hypothesis. We discuss this in paragraph 3, section 7. However, as we state in the abstract `such studies do not usually incorporate the geometry of the originating domain or task in a substantial way` beforehand, and `even fewer do so for multi-agent systems`. We ask Reviewer 2GJX to please point to the significant literature that does so, e.g., studies using positive curvature as a feature in a task representation while simultaneously studying the subsequent neural representations for similar features.
>
> **Missing technical discussion**
>
> `It is necessary to explain explicitly how generators stand for the more common notion of turn-taking. That is, the state complex also discretizes time from the perspective of agents.`
> `Also explain how the notion of commutativity relates to simultaneous action by independent agents, i.e., agents perceiving the system at a given state can decide to move simultaneously if the resulting actions commute.`

---

> > ### Comment · Reviewer_2GJX · 2023-09-27
> > **Final response to the remaining parts of the rebuttal + recommended citations**
> >
> > This is my final position on the remainder of the rebuttal. It is up to the authors to take my recommendations into account, in coordination with the action editor.
> >
> > While I appreciate the follow up from the authors, I do not have the capacity to continue to engage with messages at that length. I believe I've shared my honest impressions upon reading the paper, as a representative of the scientific community. I can only hope the authors can use my feedback to improve the submission.
> >
> > ---
> >
> > ## Follow up remarks
> >
> > I revised my account of the main result based on the author's latest comments.
> >
> > I'm afraid it does not appear further engagement in all the other points would help. Please take what I wrote as recommendations, and follow up with the action editor.
> >
> > **For future reference**, it would help to use different colors or quotation styles to distinguish what's taken from the paper, or the reviewer comment, or perhaps an author comment. It would certainly help to limit the length of any follow up comments.
> >
> > ---
> >
> > **(Recommendation):** I still recommend the authors clarify notions of time/turn-taking as commonly featured in e.g. strategy games. The author's response to this point simply quotes parts of the paper which I've already read, and which don't address that request.
> >
> > Related to that, it is still not clear to me how to reconcile the concept in "cannot be safely performed simultaneously and independently without risking collisions between labels" with the fact, confirmed by the authors, that invalid states are absent from the state graph -- which would suggest the system cannot transition into those states by definition.
> >
> > **(Recommendation):** Please clarify the definition of safety and invalid states, and what it means for the system/controller to *attempt* to apply an operation that would lead to an invalid state.
> >
> > ---
> > ## **Recommended References**
> >
> >
> > ### >> Multi-agent navigation and collision-avoidance
> > This relates to the 4th claim I called out. The account of related work presented at the top of page-2 groups current approaches into only two groups, which does not seem accurate or help with a fair positioning of the submission. As such, I noted the claim appears overstated.
> >
> > **(Recommendation)** In this very same paragraph at the top of page-2, please cite and briefly discuss (or at least acknowledge) other approaches such as:
> > - Ardila, Federico, Tia Baker, and Rika Yatchak. "Moving robots efficiently using the combinatorics of CAT (0) cubical complexes." SIAM Journal on Discrete Mathematics 28, no. 2 (2014): 986-1007.
> >   - Comment: already cited elsewhere in the paper, relies on NPC properties
> > - Arslan, Omur, Dan P. Guralnik, and Daniel E. Koditschek. "Coordinated robot navigation via hierarchical clustering." IEEE Transactions on Robotics 32, no. 2 (2016): 352-371.
> >   - Comment: also a heavily geometric approach, successfully incorporates motion models
> > - Rizk, Yara, Mariette Awad, and Edward W. Tunstel. "Cooperative heterogeneous multi-robot systems: A survey." ACM Computing Surveys (CSUR) 52, no. 2 (2019): 1-31.
> >   - Comment: it should always help to cite up-to-date surveys, especially when making broad assertions about current state-of-the-art.
> > - Verginis, Christos K., Franck Djeumou, and Ufuk Topcu. "Safety-constrained learning and control using scarce data and reciprocal barriers." arXiv preprint arXiv:2105.06526 (2021).
> >   - Comment: this is less relevant, but mainly sharing since the authors expressed interest. It appears this direction is yet to be developed, but it could be relevant here in regards to alleviating assumptions e.g. on motion dynamics.
> >
> > ---
> >
> > ### >> Impact of the geometry of the originating domain or task
> > This relates to the 6th claim I called out. The account of related work presented on the second paragraph of page-2 does not cite the most relevant papers and proceeds to present a conclusion/claim which isn't accurate and again does not help with a fair positioning of the submission. As such, I noted the claim lacks evidence.
> >
> > **(Recommendation)** In this very same second paragraph of page-2, please revise the citations to include and briefly discuss (or at least acknowledge) other research themes such as:
> > - Reda, Daniele, Tianxin Tao, and Michiel van de Panne. "Learning to locomote: Understanding how environment design matters for deep reinforcement learning." In Proceedings of the 13th ACM SIGGRAPH Conference on Motion, Interaction and Games, pp. 1-10. 2020.
> > - Peng, Xue Bin, and Michiel Van De Panne. "Learning locomotion skills using deeprl: Does the choice of action space matter?." In Proceedings of the ACM SIGGRAPH/Eurographics Symposium on Computer Animation, pp. 1-13. 2017.
> > - Schneider, Jan, Pierre Schumacher, Daniel Häufle, Bernhard Schölkopf, and Dieter Büchler. "Investigating the Impact of Action Representations in Policy Gradient Algorithms." arXiv preprint arXiv:2309.06921 (2023).

---

> > > ### Author Response · Authors · 2023-09-29
> > > **Follow-up response to ‘Final response to the remaining parts of the rebuttal + recommended citations’**
> > >
> > > Given Reviewer 2GJX states this was their `final position on the remainder of the rebuttal`, this response is mostly for the benefit of others to clarify misunderstandings and lay out our intended changes for the manuscript.
> > >
> > >
> > > **Follow up remarks**
> > >
> > >
> > > `I revised my account of the main result based on the author's latest comments.`
> > >
> > >
> > > We appreciate Reviewer 2GJX’s withdrawal from their initial assessment that computational implementations would be prohibitive for time complexity reasons. As recommended by Reviewer 2GJX, and as we describe in our follow-up responses above titled “Follow-up response 1 [and 2] to ‘Final response to part-2 of the rebuttal’”, we will add this discussion regarding computational complexity to the paper.
> > >
> > > We acknowledge Reviewer 2GJX’s request to `Please take what I wrote as recommendations` regarding all other points (which we addressed in Responses 3, 4, 5, 6). As such, all points raised under “Missing technical discussion” and “Presentation issues” shall henceforth be treated as recommendations.
> > >
> > > We note that Reviewer 2GJX also revised their assessment of our main theorem from “fairly obvious” to “appears obvious”. It is difficult to judge the scientific value of this appraisal, as it refers only to the appearance of the main result, rather than its precise formulation. As described in our follow-up responses, the result is not technically obvious and Reviewer 2GJX has not explained what makes our theorem technically obvious to them.
> > >
> > > `(Recommendation): I still recommend the authors clarify notions of time/turn-taking as commonly featured in e.g. strategy games. The author's response to this point simply quotes parts of the paper which I've already read, and which don't address that request.
> > > Related to that, it is still not clear to me how to reconcile the concept in "cannot be safely performed simultaneously and independently without risking collisions between labels" with the fact, confirmed by the authors, that invalid states are absent from the state graph -- which would suggest the system cannot transition into those states by definition.`
> > >
> > > `(Recommendation): Please clarify the definition of safety and invalid states, and what it means for the system/controller to attempt to apply an operation that would lead to an invalid state.`
> > >
> > > We have attempted to clarify these points in Response 5 (paragraphs 3, 5, and 6) and are happy to follow Reviewer 2GJX’s recommendations to add these explanations to the paper.
> > >
> > > If the agents are required to take turns to make moves (no simultaneous actions allowed), then one can simply work with the state graph. The higher-dimensional cubes allow for simultaneously valid actions to be performed at a single time step. To clarify the points regarding potential invalid states and attempting to apply operations, let us consider the following setup. Each time step involves a *planning* phase and an *execution* phase. In the planning phase, each agent may propose an action (e.g., Move). The set of proposed actions $P$ can be regarded as a set of vertices in $lk(v)$ of the current state $v$. If $P$ spans a simplex in $lk(v)$, then all actions can be safely performed simultaneously in the execution phase. If not, then we *choose* a maximal simplex in the subcomplex of $lk(v)$ spanned by $P$, yielding a maximal subset of simultaneously valid actions to be executed. More formally, we could regard such a “collision avoidance policy” as a function from the power set of $V(lk(v))$ to $lk(v)$. Thus, we can reframe the problem of collision avoidance in terms of the combinatorial structure of links which, in turn, is captured by the local geometry of the state complex.
> > >
> > > **Recommended References**
> > >
> > > We sincerely thank Reviewer 2GJX for these excellent lists of references. We are very happy to cite and discuss all of these, and agree with Reviewer 2GJX’s recommendation to do so in the second paragraph of page 2 of the paper. We note that in most of these works, where they incorporate geometric realisations, concepts, or techniques, they do so in an empirical way (whereas our contribution is more formal) and do not utilise higher dimensional objects such as simplicial or cube complexes. The exception to this is the work by Ardila et al. (2014), which as we previously discussed requires NPC as a property (whereas we use positive curvature as a condition for detecting dangerous states) and which is not applied to multi-agent navigation or collision detection. Nevertheless, by incorporating and further discussing these references in our paper, and as recommended by Reviewer 2GJX, we hope to provide a clearer and more precise context of our original contributions.

---

> ### Author Response · Authors · 2023-09-25
> **Response 4 to Reviewer 2GJX**
>
> These points are already discussed in the manuscript, e.g., in paragraph 12 of section 2 (e.g., `When [multiple generators are admissible from the same state and do not have overlapping supports], these generators can be applied independently of one another, and the resulting state does not depend on the order in which they are applied.`) and paragraphs 10-11 of section 5 (e.g., `Failure of the Link Condition can indicate available moves at some state that cannot be safely performed simultaneously and independently without risking collisions between labels.`). However, if they bear repeating or earlier mentions, we think the best place to do so would be in the first or second paragraphs of section 2.
>
> `This leaves non-commuting moves under-specified. From a physical viewpoint, those aren't realizable without collision. It is not clear if the resulting (invalid) states are included in the state graph, for otherwise the simultaneous execution of a non-commuting set of moves cannot happen by definition. For example, fewer legal actions can be selectively allowed, e.g., by tie-breaking, which seems to be how many computer games resolve this issue.`
>
> If a state is 'invalid' or not admissible, then by Definitions 2.1 and 2.2 it is not part of the state graph (nor complex, per Definition 2.3).
>
> `Please explain explicitly whether dances allow agents to move diagonally. If so, wouldn't it be more natural to include diagonal moves from the get-go? It appears dances only allow diagonal moves if the move can be executed both as a sequence of horizontal+vertical or vertical+horizontal moves, i.e., if only one of those two sequences are admissible, the diagonal move cannot happen. I don't see a justification for this restriction, since the stated purpose of dances is to more faithfully describe the continuum of a 2D plane -- following the same reasoning at the bottom of Page 6.`
>
> A dance is not a generator; rather, it is a set of 4 states related by 4 Move generators. So it does not make sense to ‘apply’ a Dance as one would with a Move. It is correct that a dance can only exist if a ‘diagonal’ move can be executed by performing horizontal+vertical or vertical+horizontal Moves in both orders. What the dance actually captures is what we might call a “spatially commuting” relationship between these two move sequences. Note that these Moves are not commuting in the strict AGP sense (which could be described as “temporally commuting”), however, our modifications allow us to treat both notions of commutativity simultaneously via cubes. This demonstrates a conceptual advantage of working with a complex rather than a graph: with a graph, we can only capture relations between pairs of states; by allowing higher-dimensional cubes, we can also capture relations between sets of generators/states (e.g., commutativity or dances).
>
> One could certainly choose to add edges to the state complex to represent diagonal moves instead of filling in ‘dance’ squares. However, doing so would further complicate the topology of the state complex. In the example of a 2x2 gridworld with one agent, we would end up with a complete graph on 4 vertices as the state graph/complex. This is homotopy equivalent to a wedge of 3 circles, hence its fundamental group is a free group of rank 3. However, the original state complex is homotopy equivalent to a circle, with the group of integers (the free group of rank 1) as its fundamental group. Heuristically, adding more edges would increase the rank of the fundamental group, introducing even more unnatural topology. To reduce the rank, we should attach higher dimensional cells bounded by loops - this has the effect of forcing the corresponding group elements to be trivial. So, by filling in ‘dance’ squares, the modified state complex in the 2x2 example becomes contractible, and hence has trivial fundamental group. If one still insists on having edges for diagonal moves, we could fill in triangles or tetrahedra whenever they appear in order to remove unnatural topology. However, the resulting complex would contain a mix of cubes and tetrahedra. We do not feel this would yield a more natural object than the one we proposed.
>
> Background on fundamental groups and homotopy can be found in standard topology texts: see Hatcher (2002) or [Mun00].

---

> ### Author Response · Authors · 2023-09-25
> **Response 5 to Reviewer 2GJX**
>
> `Please note that the article makes no mention of CAT(0).`
>
> For a cube complex X, the CAT(0) property is equivalent to X being NPC and simply connected (i.e. it has trivial topology). The AGP state complexes always satisfy NPC, but they are not necessarily simply connected. For example, the state complex for the 2x2 gridworld with 2 agents is homotopically equivalent to a circle, hence it is not simply connected, and therefore not CAT(0). Thus, we do not mention CAT(0) because it would be technically incorrect. We could mention that “NPC” is synonymous with “locally CAT(0)” for the benefit of readers who may have come across CAT(0) geometry. In the work of Ardila et al. on robotic arms, it is simple connectedness (a global property) which is harder to establish; the NPC property comes for free from AGP.
>
> `similar formulations were studied in prior works, for example: Ghrist, Robert, and Steven M. Lavalle. "Nonpositive curvature and pareto optimal coordination of robots." SIAM Journal on Control and Optimization 45, no. 5 (2006): 1697-1713. Please cite and discuss, clearly positioning the proposed contributions in this light.`
>
> We thank Reviewer 2GJX for bringing this article to our attention. The ‘roadmap co-ordination space’ defined by Ghrist-Lavalle is constructed by taking a product of $n$ graphs, then deleting an ‘obstacle’ set. Here, each graph involved represents a ‘roadmap’ where the corresponding agent/robot can move. We can adapt this to the setting of $n$ (distinguished) agents in a gridworld: choose the gridworld as the roadmap for each agent, then delete cubes when there is a collision between two agents (this needs to be done carefully). We expect the resulting roadmap co-ordination space to be the same as the state complex for the (labelled) $n$ agent gridworld obtained by directly using the AGP setup. In other words, both approaches should yield the same object in the context of distinguished agents in a gridworld. Under either approach, one obtains an object which is NPC.
>
> The roadmap configuration space / AGP state complex frameworks suffer from a common limitation: each agent is confined to move on a graph – a $1$-dimensional space. However, a gridworld ought to be regarded as a discretized $2$-dimensional domain. In this light, we could try defining a ‘modified roadmap configuration space’ which allows each roadmap to be a (discretized) rectangle (or whatever shape the room might be). It seems plausible that by deleting ‘collision’ cubes in an appropriate manner, one should recover our modified state complex (for $n$ labelled agents). If so, it would further support the case that filling squares whenever there are dances is the most natural approach. Indeed, our guiding intuition for the modified state complex of an $n$-agent gridworld is that it should serve as a discrete analogue of the configuration space of $n$ points on the plane.
>
> `Would it help to use computer-aided proofs or enumeration to characterize NPC failures involving objects or multiple agents+objects?`
>
> In order to address this, one must first decide what higher dimensional cubes ought to be added in order to define a ‘natural’ modified state complex involving objects. Once the ruleset/cubes have been decided, enumeration of all possible links, as we have done in the agent-only case, would be an obvious first approach to characterizing NPC failures in more general settings. Computer-aided methods could be helpful in the situation where the number of cases becomes large, as it should not be difficult to extend the link computation function in our Python tool to work with more general gridworlds. Nevertheless, it would be nice to obtain a characterization of NPC failures without the need for enumeration, as this would provide greater mathematical insight.
>
> **Presentation**
>
> `The notation $S^{(1)}$ was not justified.`
>
> For a cell complex X, it is standard notation to use $X^{(k)}$ for its $k$-skeleton (the subcomplex formed by the union of all cells of dimension at most k). Since the state graph is the 1–skeleton of the state complex S, this justifies the notation $S^{(1)}$ in Definition 2.2.
>
> `The discussion (3 paragraphs) near Figure 3 seems too verbose, and seems to reappear each time a complex is defined.`
>
> It is unclear which 3 paragraphs Reviewer 2GJX  is referring to. However, we view most if not all of the text on page 3 (the page which Figure 3 is on) to be essential in sufficiently setting up the mathematical concepts, framework, and notation which we rely on in later sections. We also could not identify which text is repeated, nor any repetitions of the state complex definition (Definition 2.3).
>
> `In justifying small environments (first paragraph of Section 6) I found the analogy to integer addition unnecessary. The remark was much more relevant to the discussion.`
>
> We propose to move this to the discussion (or a new appendix section) to more deeply discuss the computational aspects.

---

> ### Author Response · Authors · 2023-09-25
> **Response 6 to Reviewer 2GJX**
>
> `It is not immediately clear how label-inversion and loss of symmetry help the main development, or fit the same motivations towards practical applications.`
>
> The presence of the label-inversion symmetry, which occurs if one directly uses the AGP setup, indicates that ‘agent’ and ‘empty’ labels perform equivalent roles from the viewpoint of the reconfigurable system. Our modifications (adding cubes for agent dances) breaks this symmetry, and thus distinguishes these two roles. From a practical perspective, we can then use failure of NPC in our modified state complexes to detect a potential collision of agents, rather than a potential ‘collision’ of empty cells.
>
> `limiting the study to connected state graphs makes this less interesting`
>
> We agree that there is no theoretical reason for limiting our study to connected state complexes. In fact, our main theorem would still hold without the connectedness assumption as it is a purely local statement. If one begins with an initial state, then its connected component comprises precisely all states which can be reached by applying some sequence of generators. Thus, from a practical and computational standpoint, one only ever deals with a connected component of the full state complex when exploring from a specific instance of a gridworld. Another issue is that without the connectedness assumption, the problem of finding a shortest path between two given states could be impossible.
>
> `Section 3 appears largely unrelated to the development towards the main result, and does not offer specific technical contributions.`
> `section [3] seems to report a collection of observations before digressing into further unanswered questions.`
>
> Section 2 describes the mathematical framework and introduces key definitions. Section 3 then describes how we compute and visualise state complexes of gridworlds, and highlights what can be learnt about the intrinsic geometry and topology of gridworlds by doing so. For example, this section contains and discusses two canonical examples: the first, a $3 \times 3$ gridworld with one agent and one object, demonstrates different scales of geometry of the task embedded in the complex; the second, a $3 \times 3$ gridworld with three or six agents, demonstrates how a symmetry between different gridworlds can generate an identical underlying configuration space (which in this example is no longer the case after our later modification), and a state complex which is non-planar. By developing these and other intuitions, this section is also designed to help support and lead the reader towards the arguments in Section 4, where we introduce our modification.
>
> `It may be true that visualizations can be revealing for small subsets of the state graph, but this point was not adequately developed.`
>
> Contrastingly, Reviewer oPeG said `good examples and illustrations are given` and Reviewer jxVD commended our `concrete illustrative experiments`. We believe the provided visualisations (shown and discussed at length for many, complete state complexes, especially in Section 3 and in Appendix A1) are helpful and necessary guides which develop readers’ intuitions about the underlying mathematics and system.
>
> `Last paragraph of Section 4 seems to digress beyond the scope of the paper`
>
> The last paragraph of Section 4 was intended to briefly hint at our true topological motivations for filling in cubes associated with dances: to create a discrete analogue of a configuration space of $n$ (unlabelled) points on the plane. However, we did not want to assume that readers had the necessary mathematical background, so we chose not to elaborate on this point too much in the current paper. Nevertheless, for those familiar with configuration spaces and braid groups, it should be reasonably clear what we are trying to do.
>
> This paragraph also makes the important point that certain symmetries in the unmodified state complex, like shown in Figure 5, become lost, since there is no ‘inverted dance’, i.e., a support with three agents and one empty space. This nicely concludes the entire section, which lays out how we compute and visualise state complexes, and develops interesting geometric insights and intuitions relied on in later sections.
>
> **Additional References**
>
> [Bel21] Mark Bell. bigger (Computer Software). pypi.python.org/pypi/bigger, 2021. Version 0.3.1.
>
> [BKL98] Joan Birman, Ki Hyoung Ko, Sang Jin Lee. A New Approach to the Word and Conjugacy Problems in the Braid Groups. Adv Math 139, 322-353 (1998).
>
> [FG03] Nuno Franco, Juan González-Meneses. Conjugacy problem for braid groups and Garside groups. J Algebra 266, 112-132 (2003).
>
> [GL06] Rob Ghrist and Steven M. Lavalle. Nonpositive curvature and pareto optimal coordination of robots. SIAM Journal on Control and Optimization 45(5), 1697-1713 (2006).
>
> [Mun00] James Munkres. Topology, Second Edition (2000).

---

### Review · Reviewer_jxVD · 2023-09-10

**Summary Of Contributions:**

The authors takes the theory of state complexes in geometric group theory to multi-agent grid-worlds.
They provide a well-written introduction to this area, which is quite far removed from almost all work in the multi-agent reinforcement learning field to whom I take this as being targeted.
The authors noticed a connection between the geometric concept of positive curvature in the state complex and the risk of collision between agents moving in this world.

**Audience:**

Yes

**Claims And Evidence:**

Yes

**Requested Changes:**

I do not have any concrete input to this work, which I review with very low confidence.

**Strengths And Weaknesses:**

It well-written and does make me interested in learning more about state complexes to see what it can add to how I think about tasks I study, though it is far from clear to me at this point even if I find the connections intriguing.

The main weakness is that the article targets an audience and journal for whom the material is far removed and what it delivers is not really tangible but still seems more of theoretical mathematical interest in the sense of adding more examples of interest to the mathematical study of state complexes and ways general geometric properties materialises in them. The authors do, however, make attempts at alleviating this weakness by some concrete illustrative experiments. The direction of seeing if this theory can help bring understanding to what representations are learnt by deep neural networks I find the most promising for further future work.

---

> ### Author Response · Authors · 2023-09-25
> **Response to Reviewer jxVD**
>
> We thank Reviewer jxVD for their supportive review. Indeed, we expect some of the mathematical concepts used in this paper will be new to readers in the machine learning community. We have therefore made significant attempts to include concrete, illustrative examples and experiments (as Reviewer jxVD notes), ensure the content and concepts are clearly explained for an audience without as much formal training in these areas (as noted by Reviewer oPeG), and provide “a friendly introduction” (as noted by Reviewer 2GJX).
>
> We are happy to hear that, upon reading, Reviewer jxVD became “interested in learning more about state complexes to see what it can add to how I think about tasks I study” and concluded their review with “[t]he direction of seeing if this theory can help bring understanding to what representations are learnt by deep neural networks I find the most promising for further future work.” These are exactly the kind of main messages we hope readers of the paper will come away with! We, too, are excited for the future work which studies, and incorporates in a fundamental way, the geometry and topology of tasks in ML and AI.

---

### Review · Reviewer_oPeG · 2023-09-14

**Summary Of Contributions:**

This paper provides an algebraic/topological analysis of a simple grid-world set up used in robotics and AI.  It considers a fixed grid-defined state space where each position can be occupied by a single agent or a single object, or not occupied.  Agents can move to a neighboring unoccupied cell, and can push or pull adjacent objects with them.

The papers main results is an extension of a previous characterization by Adams, Ghrist, and Peterson.  The new analysis identifies some rare situations which do not satisfy the conditions from the prior work which ensures the space is non-positively curved.  These cases correspond with potential grid lock and other challenges in planning.

While this paper does not directly use these findings in AI for robotics, they could potentially provide the scaffold for others to do so, in ways that the work of Adams, Christ, and Peterson did in years before.

**Audience:**

Yes

**Broader Impact Concerns:**

No broader impact concerns.

**Claims And Evidence:**

Yes

**Requested Changes:**

The paper is fine for what it is.

**Strengths And Weaknesses:**

Strengths:
 - the writing is clear and concise, and translates complex math in a way that could be accessible for someone without as much formal training in that area
 - good examples and illustrations are given

Weaknesses:
 - the work analyzes a simplistic model that does not capture many complex motion planning.  Yet, this sort of set-up is used in many planning areas of robotics when the world can indeed be discretized.
 - the main contribution is a bit hard to understand, and one basically needs to read the entire paper to understand what the contribution is.
 - the paper does not make connections directly into ML/AI, but rather provides mathematics that could be used as the basis of contributions.

---

> ### Author Response · Authors · 2023-09-25
> **Response to Reviewer oPeG**
>
> We thank Reviewer oPeG for their thoughtful review. We particularly appreciate the following comment: “While this paper does not directly use [its] findings in AI for robotics, they could potentially provide the scaffold for others to do so, in ways that the work of A[brahms], [G]hrist, and Peterson did in years before.” Reviewer jxVD echoed a similar potential for future applications: “The direction of seeing if this theory can help bring understanding to what representations are learnt by deep neural networks I find the most promising for further future work.”
>
> Overall, while our paper might be more mathematical than some TMLR papers, we are thankful all three reviewers recognised the effort we put into the presentation of the content and concepts for readers without as much formal mathematics background:
> - Reviewer oPeG: “the writing is clear and concise, and translates complex math in a way that could be accessible for someone without as much formal training in that area”
> - Reviewer jxVD: “It [is] well-written and does make me interested in learning more about state complexes to see what it can add to how I think about tasks I study”
> - Reviewer 2GJX: “the paper offers a friendly introduction to the construction of state complexes for simple environments with agents and movable objects”

---

### Comment · Action_Editors · 2023-10-12
**Update paper for final recommendation?**

Dear Authors.

Often TMLR authors update the paper in response to reviewer discussion during the discussion phase.  I admit this part of the process is not totally clear -- and is not strictly necessary either.

It is now time for the reviewers to make a final recommendation (they have 2 weeks).  Especially in light of the long and detailed discussion with Reviewer 2GJX, it would be useful to see a few of the recommendations discussed seen in place in the paper to aid in the reviewer recommendations.  From my reading of the discussion, it seems that ultimately, most of the changes would involve including some discussion of related work, clarifying the distinction between some technical terms, and potentially rephrasing some descriptions of the results.  I think reviewers would find it useful to see this put into action, and a short comment from the authors of what was changed.

thanks!

---

> ### Author Response · Authors · 2023-10-13
> **Intended revisions**
>
> We are very happy to incorporate some recommendations in the form of a revision, and fully expected to do so for the final version. We were not expecting to do so during the discussion period. Some of these revisions require careful crafting and reworking of the text to ensure clarity and avoid further confusion. Some of this work has already been done in the replies, and we intend to re-use parts of those replies in the revised version. However, our current schedules mean we are unable to guarantee that we can provide such a revision within the next two weeks.
>
> Based on the discussion, and as indicated in our replies to reviewers, we intend to:
> 1. Add discussion of prior work on state graphs and emphasize the distinction between graphs vs complexes in the introduction section.
> 2. Add the computational discussion to section 6.
> 3. Make it clear in the introduction and abstract that our proof is for the agent-only case.
> 4. Potentially repeat earlier on (e.g., beginning of section 2) how the discretization works, or perhaps better is we add some discussion of "temporal commutativity" in section 2 and "spatial commutativity" in section 5.
> 5. Include a finer discussion on the interpretation of collision-detection to section 5.
> 6. Cite and discuss the recommended references provided by Reviewer 2GJX.

---

### Decision · Action_Editor_u4a5 · 2023-10-29

**Recommendation:** Accept with minor revision

**Comment:**

This is a fairly specialized result, but should prove relevant for planning problems in AI, which is within TMLR's scope.  It provides mathematical foundation for an actively used class of planning problems, and characterizes a set of scenarios which require specific attention in planning.
This will help formalize when certain planning approaches will work, and identify specific scenarios where things will not automatically work, and special cases will be needed to handle.

However, there are numerous things that should be addressed before the paper is accepted.  These will need to be reviewed by the action editor (myself) before final approval.
With regard to this, I approve the authors plan (amended as):
> 1. Add discussion of prior work on state graphs and emphasize the distinction between graphs vs complexes in the introduction section.
> 2. Add the computational discussion to section 6.
> 3. Make it clear in the introduction and abstract that our proof is for the agent-only case.
> 4. Add discussion of "temporal commutativity" in section 2 and "spatial commutativity" in section 5.
> 5. Include a finer discussion on the interpretation of collision-detection to section 5.
> 6. Cite and discuss the recommended references provided by Reviewer 2GJX.

Please provide a summary of changes when you have made them.

**Audience:**

All reviewers agreed that it satisfied the "audience" criteria.  I do think these results are somewhat niche for the TMLR audience, but I do think it will find an interested audience.

**Claims And Evidence:**

All reviewers agreed that it satisfies the "claims and evidence" criteria.  One reviewer has significant concerns about the framing of this paper with respect to related work.  I think this can be addressed in minor revisions, and these will need to be shepherded by the AE (myself).

---

> ### Author Response · Authors · 2023-11-28
> **Revision submitted**
>
> We wish to thank the Action Editor for their considered decision and in advance for shepherding these final minor revisions. Thank you, also, to the reviewers for their diligent work in offering feedback (especially Reviewer 2GJX), which we have now incorporated to improve the paper.
>
> Following-up from our planned changes, we have:
> 1. Added discussion of prior work on state graphs (added paragraph 2 and modified the final paragraph of the introduction) and emphasised the distinction between graphs vs. complexes in the abstract and introduction.
> 2. Modified sentences in the introduction and abstract to make it clearer to readers that our proof is for the agent-only case.
> 3. Added footnote 1 (in Definition 2.2) to explain the notation. Added footnote 3 in the beginning of Section 4 to justify and explain why we limited the study to connected state complexes.
> 4. Added paragraph 5 in Section 4 to describe how the dances capture the notion of ‘spatially commuting’ moves. (We chose not to add a discussion of ‘temporally commuting’ to Section 2 as it made more sense to mention it here.) Slightly reworded the first sentence of the next paragraph.
> 5. Added Remark 4.2 to explain why we chose to fill in squares for dances instead of simply adding edges for diagonal moves.
> 6. Added Remark 5.1 to explain that the terms ‘NPC’ and ‘locally CAT(0)’ are synonymous. Also explained that the stronger CAT(0) property does not hold in general for state complexes (as it further requires simple connectedness).
> 7. Added paragraph in Section 5 (two paragraphs above Theorem 5.2) to further explicate the connection between links and collision avoidance. Slightly reworded first sentence of following paragraph.
> 8. Added paragraph in Section 6 (two paragraphs below Remark 6.1) to quantify the reduction in computational workload in computing links due to the main theorem.
> 9. Cited and discussed all papers suggested by Reviewer 2GJX in the introduction and/or discussion sections.

---

> > ### Comment · Action_Editors · 2023-12-06
> >
> > Thanks to the authors for the careful revision.  I believe all concerns have been addressed, and the paper is ready for publication.